# Separator with high ionic conductivity enables electrochemical capacitors to line-filter at high power

Yajie Hu [1], Puying Li[1], Guobin Lai[2,3], Bing Lu[1], Haiyan Wang[1], Huhu Cheng [1], Mingmao Wu [2], Feng Liu [3], Zhi-Min Dang [4] & Liangti Qu [1] ✉

Line-filtering electrochemical capacitors (LFECs) are demonstrating advantages in line filtering over traditional electrolytic capacitors. However, they can only function at no-load or low-power conditions due to the limited high-frequency capacitance resulting from the excessive ionic resistance, despite much progress in electrode materials. Here, we show separators dominate both ion migration and capacitance in LFECs. A 3 μm-thick thread-anchor structured separator is developed, featuring both accelerated ionic transport and reliability, leading to a low ionic resistance of 25 mΩ cm². With a phase angle of −80° at 120 Hz, the assembled device has an areal capacitance of 6.6 mF cm⁻². Furthermore, stack integration in parallel breaks the trade-off between capacitance and frequency response, boosting the areal capacitance by two orders of magnitude without decay of frequency characteristics. The On-board field test demonstrates that voltage ripples are steadily suppressed below 5% even for practical high-power line filtering with a load power density of 2.5 W cm⁻², three orders of magnitude higher than previous instances. This work opens up a perspective of separator engineering for the development of high-performance line-filtering electrochemical capacitors and promotes their applications in practical high-power scenarios.

AC line-filtering capacitors smooth ripples and noises out of the power signals to secure the normal operation of appliances, especially for alternating current to direct current conversion. However, to meet the typical frequency characteristics[1,2] (phase angle, $\varphi < -80°$ at 120 Hz), AC line-filtering capacitors are, at present, mainly electrolytic capacitors, whose cumbersome sizes and excessive footprints severely restrict the circuit miniaturization[3,4].

Line-filtering electrochemical capacitors (LFECs) theoretically possess four orders of magnitude higher specific capacitance than electrolytic capacitors[5], deemed as the next-generation candidate[6,7]. Nonetheless, restricted by excessively high series resistance (*SR*) at 120 Hz, the capacitance faces severe compromise in exchange for frequency responses. Thus, the practical implementation of LFECs is only demonstrated feasible at low-power or no-load conditions[8–13]. For practical power filtering with a load power density (the supplied power divided by the device footprint) higher than 10 mW cm⁻², the ripple factor surges to more than 10%, far beyond the norm of practical line filtering (Supplementary Table 1).

Efforts have been devoted to improving capacitance by fabricating thick electrode materials with straight-trough pores, as the inferior frequency responses are prevalently thought to originate from the ionic impedances in electrode materials[1,2,14,15]. However, the thickness

[1]Key Laboratory of Organic Optoelectronics & Molecular Engineering, Ministry of Education; State Key Laboratory of Flexible Electronics Technology; Department of Chemistry, Tsinghua University, Beijing 100084, P. R. China. [2]Key Laboratory of Eco-materials Advanced Technology, College of Materials Science and Engineering, Fuzhou University, Fuzhou, Fujian 350108, P.R. China. [3]State Key Laboratory of Nonlinear Mechanics, Institute of Mechanics, Chinese Academy of Sciences, Beijing 100190, P. R. China. [4]State Key Laboratory of Power System Operation and Control, Department of Electrical Engineering, Tsinghua University, Beijing 100084, P. R. China. ✉e-mail: lqu@mail.tsinghua.edu.cn

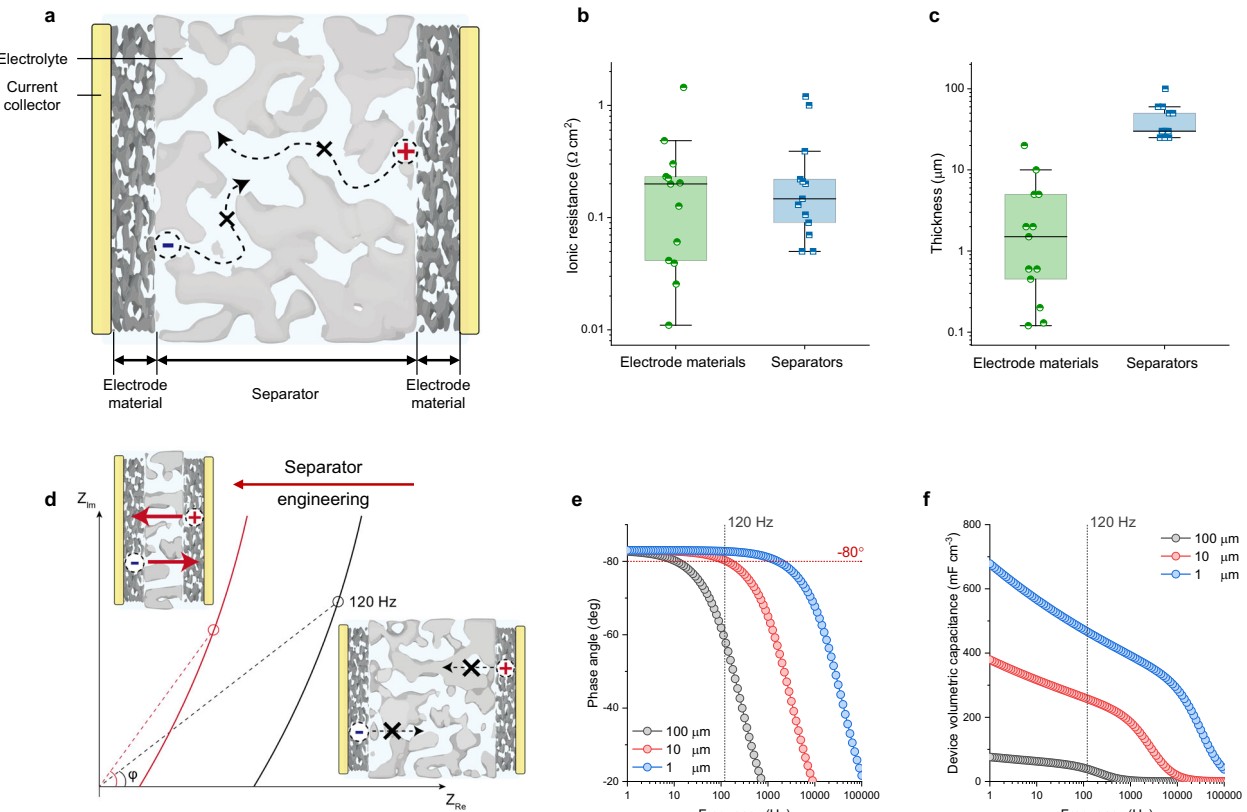

**Fig. 1 | Design principle of TAS-LFECs. a** Schematic diagram of conventional LFECs and the constitution of ionic resistance. The dashed arrow indicates the sluggish ionic migration in separators. **b** Box plot of ionic resistance of electrode materials and separators in conventional LFECs (center line, median; box limits, upper and lower quartiles; whiskers, 1.5× interquartile range). See details in Supplementary Table 2. **c** Box plot of thicknesses of electrode materials and separators in conventional LFECs. See details in Supplementary Table 2. **d** Schematic Nyquist diagram showing the theoretical change of LFECs after the reduction of separator thickness. The arrows indicate ion migration. **e** The calculated phased angle at 120 Hz of LFECs with different separator thicknesses. **f** Calculated device volumetric capacitance at 120 Hz of LFECs with different separator thicknesses.

increment and tortuosity decrease are mutually repulsive, making it rather tricky for electrode material engineering[1,16,17]. Consequently, for LFECs reported so far, $SR$ at 120 Hz generally remains larger than 100 mΩ cm² with a bottleneck areal capacitance ($C_A$) of ~3 mF cm⁻² (while $\varphi < -80°$)[1,12,18,19].

Separators are the core component of sandwich-type LFECs but have gained little attention to date (Fig. 1a). We conducted a statistical analysis of the separator parameters from previous reports. Surprisingly, ionic resistance contributed by separators accounts for 54% of total $SR$ (Fig. 1b), evincing that ions are much hindered in separators. Meanwhile, commercial separators are five times thicker than electrode materials and occupy excessive non-energy-storage spaces, rendering low specific capacitance of the device (Fig. 1c). In this regard, refinement on separators should be of significance for further reduction of $SR$ and improvement of specific capacitance (Fig. 1d).

Theoretical analysis corroborates our hypothesis (Supplementary Note 1). Assuming that the separator remains mechanically and chemically stable, the reduction of separator thickness could drastically reduce $SR$ by an order of magnitude. With separator thickness varying from 100 μm to 1 μm, the phase angle at 120 Hz is calculated to be promoted from −60° to −82.8° (Fig. 1e); the device volumetric capacitance is concurrently improved by a factor of 10 (Fig. 1f). Nonetheless, there is no conventional separator that can reach such a low ionic resistance required by line filtering (< 50 mΩ cm²) while retaining stability[20].

Herein, we develop a highly ion-conductive thread-anchor structured separator (TAS) based on the interactions between cellulose nanofibers (CNFs) and graphene oxide (GO) nanosheets. The thread-anchor architecture combines abundant macropores with strong

mechanical properties (Fig. 2a). By conducting TAS-enabled 3-dimensional stack-type integration in parallel, $C_A$ is improved largely without compromising frequency responses, effectively solving the dilemma of thickening electrode materials. On this basis, high-power line filtering is accomplished with steady ripple suppression and thermal control.

## Results

### Preparation and characterization of TAS

We developed a size-matching time-lapse assembly method for the fabrication of TAS (see Methods). Firstly, sub-micrometer GO nanosheets are prepared to conform to the CNF scale (Supplementary Fig. 1), and the C/O atomic ratio is controlled at around 2.1 for the electronic insulation and interaction with CNFs. The two components are assembled in a highly diluted condition, as traced by Zeta potential change and size distribution (Supplementary Fig. 2). After filtration, the wet film is processed by time-lapse evaporation and high-temperature vacuum drying successively to release internal stress[21] and reinforce the binding interactions[22].

The as-prepared free-standing TAS is remarkably durable, capable of being bent over 180° without cracks (Supplementary Fig. 3), despite its thickness as low as 3 μm (Fig. 2b). This is attributable to the unique thread-anchor microstructure, in which GO nanosheets act as anchors to bridge the nearby CNFs (threads), forming uniform macropores of hundred nanometers, as illustrated by scanning electron microscopy (SEM) (Fig. 2c and Supplementary Fig. 4). Validation is also provided by high-angular annular dark-field scanning transmission electron microscopy (Fig. 2d). The bright regions with high contrast reflect the combination between CNFs and GO nanosheets. Besides, atomic force

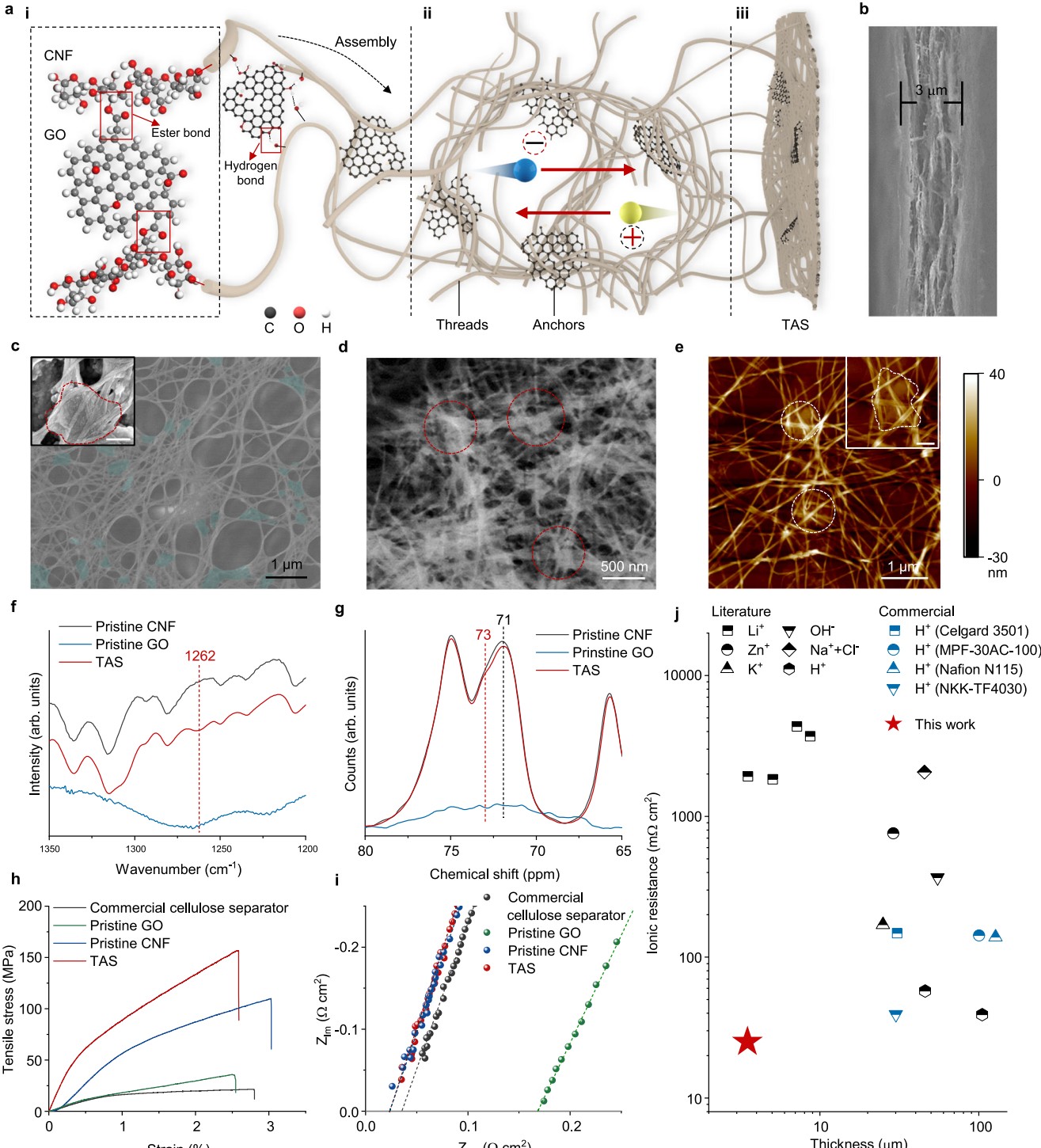

**Fig. 2 | Characterization of TAS. a** Schematic of TAS, including the interactions between CNFs and GO nanosheets (**i**), the assembly of the thread-anchor microstructure (**ii**), and the TAS (**iii**). Inset is the calculated atomic structure of CNFs and GO nanosheets in TAS, showing the covalent ester bonds thereof (Supplementary Data 1). The red arrow refers to the migration of ions. **b** Lateral-view scanning electron microscopy (SEM) image of TAS. **c** Top-view false-color SEM image of the thread-anchor structure of TAS. The turquoise-colored regions indicate GO anchors. Inset is the magnified SEM image. Scale bar in the inset, 200 nm. The red dashed circle outlines the GO anchor. **d** High-angle annular dark-field scanning transmission electron microscopy (HAADF-STEM) image of the thread-anchor structure of TAS. **e** Atomic force microscopy (AFM) images of the thread-anchor structure of TAS. The white dashed circles indicate GO anchors. Inset is the high-magnification AFM image. Scale bar in the inset, 300 nm. The white dashed circle outlines the GO anchor. **f** Infrared spectra of pristine CNF, pristine GO, and TAS. **g** $^{13}$C solid-state nuclear magnetic spectra of pristine CNF, pristine GO, and TAS. **h** Stress-strain curves of commercial cellulose separator (NKK-TF4030), pristine GO, pristine CNF, and TAS. **i** Nyquist diagram of commercial cellulose separator, pristine GO, pristine CNF, and TAS in the electrolyte of 3 M H$_2$SO$_4$. **j** Comparison of ionic resistance and thickness between TAS and other commercial separators and in-literature separators. The legends indicate the separators with dominant conductive ions, including lithium ions[36–39], zinc ions[40], potassium ions[41], hydroxide ions[42], sodium and chloride ions[43], and hydrogen ions[44,45].

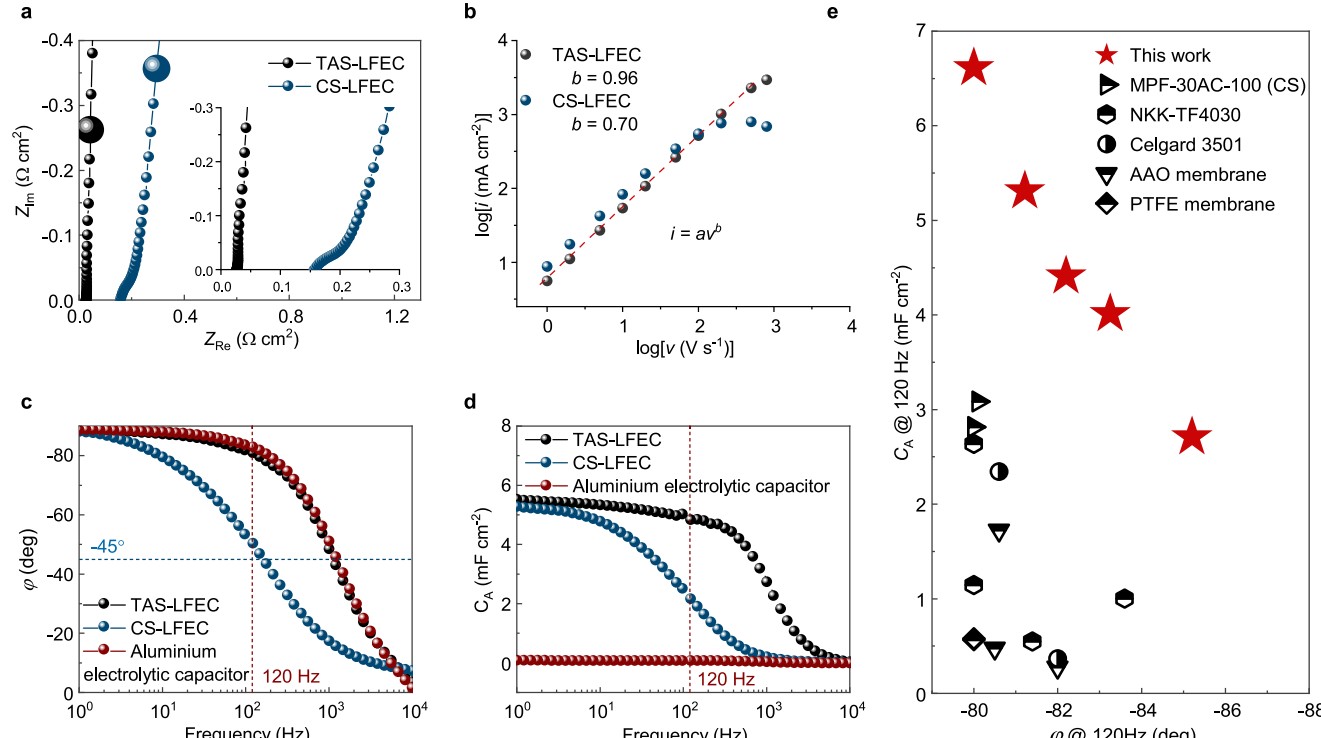

**Fig. 3 | Electrochemical performances of TAS-LFECs. a** Nyquist diagram of CS-LFEC and TAS-LFEC. Inset is the high-frequency part of the Nyquist diagram. The enlarged points indicate the impedances at 120 Hz. **b** Plot of the logarithm of current density ($i$) versus the logarithm of scan rates ($v$) for CS-LFEC and TAS-LFEC. The red dashed line is the fitting curve, and the $b$ value is determined from the slope of the curve. **c** Bode plots of commercial aluminum electrolytic capacitor, CS-LFEC, and TAS-LFEC, respectively. **d** Plots of $C_A$ versus frequency of commercial aluminum electrolytic capacitor, CS-LFEC, and TAS-LFEC, respectively. **e** Comparison of $\varphi$ and $C_A$ at 120 Hz between TAS-LFEC with other LFECs based on different separators, including MPF-30AC-100[1,46], commercial cellulose separator[11,18,23,24], Celgard 3501[12,47], AAO membrane[14,15,48], PTFE membrane[49,50]. See details in Supplementary Table 3.

microscopy (AFM) shows that GO nanosheets are embedded amidst CNFs, verifying their effective combination (Fig. 2e and Supplementary Fig. 5).

Spectroscopic measurements provide statistical verification for the assembly of CNFs and GO nanosheets. Raman spectrum of TAS shows both the specific peaks of CNFs and GO nanosheets, proving their coexistence (Supplementary Fig. 6). The D to G band ($I_D/I_G$) ratio changes from 1.86 to 1.67, mainly caused by hydrogen or covalent bonding between GO and CNF. X-ray photoelectron spectroscopy also introduces evidence of covalent bond formation (Supplementary Fig. 7). Both in O 1$s$ and C 1$s$ spectra, the peaks corresponding to O−C = O bonds (541 eV and 288.7 eV separately) strengthen slightly, in tandem with the peak attenuation of O−C bonds (532.8 eV and 286.5 eV separately). Moreover, in the infrared spectrum (Fig. 2f and Supplementary Fig. 8), a newly emerged peak at ~1262 cm$^{-1}$ is observed, corresponding to the C−O−C in the ester bond, consistent with the previous reports of GO esterification[22]. Meanwhile, the solid-state nuclear magnetic resonance spectrum of TAS reveals a subtle new shoulder peak at 73 ppm, indicating some C−O bonds have changed their chemical environment (Fig. 2g and Supplementary Fig. 9). All the evidence points toward the covalent interactions between CNFs and GO nanosheets, enabling the stability of the thread-anchor structure.

As shown in Fig. 2h, TAS exhibits a tensile strength of up to 156 MPa, far higher than the prevalently used commercial cellulose separator (NKK-TF4030)[1,11,18,23], and over 40% higher than the pristine CNF membrane. Fortunately, the mechanical strengthening doesn't compromise the ionic conductivity. Verified by electrochemical impedance spectroscopy (EIS), the ionic resistance of TAS comes down to 25 mΩ cm$^2$ in the electrolyte of 3 M H$_2$SO$_4$. This value is almost identical to the pristine CNF membrane and significantly lower than the resistance of the commercial cellulose separator (39 mΩ cm$^2$)

(Fig. 2i). Note the commercial cellulose separator cannot withstand high acidity and dissolves within 10 min, while TAS can remain stable with no degradation being observed over 10 days (Supplementary Fig. 10). Compared with previously reported and commercially used separators, TAS possesses the minimum thickness and ionic resistance, showcasing an order of magnitude reduction (Fig. 2j).

**Electrochemical performances of TAS-LFECs**
To investigate the electrochemical performances, TAS is assembled into a sandwich-type electrochemical capacitor (TAS-LFEC), with a typical composite electrode material we reported previously (Supplementary Figs. 11–14)[3,24,25]. For comparison, we choose a prevalently used commercial separator (NKK-MPF30AC-100, denoted as CS) as control.

The Nyquist curve of CS-based electrochemical capacitors (CS-LFECs) shows a large $SR$ of 295 mΩ cm$^2$ at 120 Hz and transforms a bit twisted at the high-frequency region due to the hindered ionic transport. In contrast, TAS-LFECs demonstrate a linear Nyquist curve nearly perpendicular to the abscissa axis, evincing the typical capacitor-type responses (Fig. 3a). Benefiting from the high ionic conductivity of TAS, $SR$ at 120 Hz is reduced down to 36 mΩ cm$^2$ in TAS-LFECs, minimum among previous studies (Supplementary Fig. 15 and Supplementary Table 3). Besides, a kinetic factor $b$ is derived from the cyclic voltammetry curves at scan rates ranging from 1 V s$^{-1}$ to 1000 V s$^{-1}$ (Fig. 3b and Supplementary Fig. 16). $b$ value approaching 1 indicates the kinetic behavior is like a capacitor other than an ion-diffusion-regulated battery[26]. $b$ of TAS-LFECs is measured to be 0.96, verifying their good rate characteristics, while $b$ of CS declines to 0.70 because of the severe polarization in cyclic voltammetry curves.

As expected, TAS-LFECs have better filtering-related electrochemical performances than CS-LFECs. When the electrode material

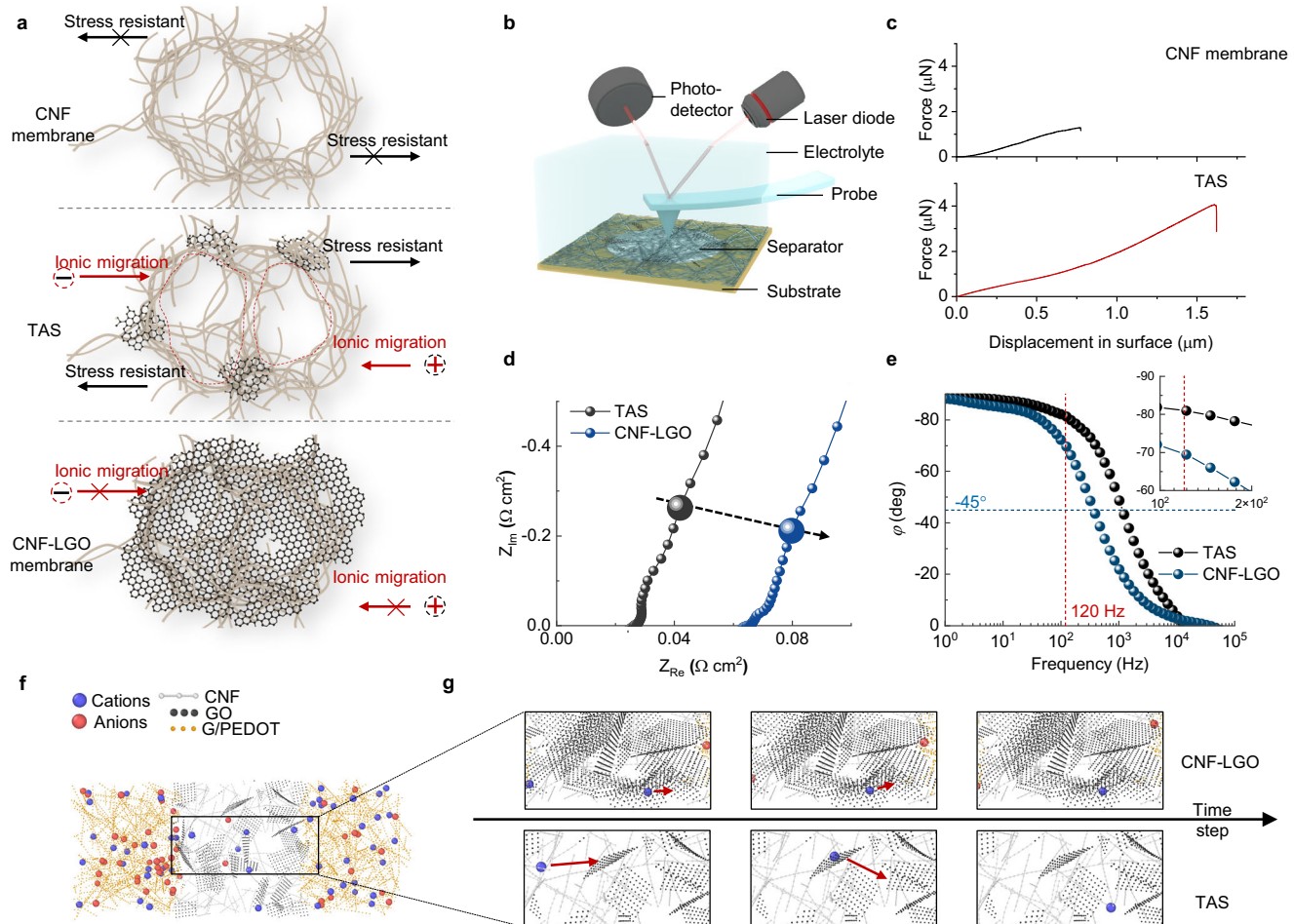

**Fig. 4 | Mechanisms underlying performance enhancement. a** Schematic diagram of the microstructure of pristine CNF membrane (upper panel), TAS (middle panel), and CNF-LGO membrane (bottom panel). **b** Schematic diagram of the in-electrolyte nano-indentation experiment. **c** Plots of force versus displacement in the surface of the CNF membrane and TAS at the in-electrolyte condition. **d** Nyquist diagrams of TAS-LFEC and CNF-LGO membrane-based LFEC. The enlarged points indicate the impedances at 120 Hz. The dashed arrow indicates the tendency for variation in *SR*. **e** Bode diagrams of TAS-LFEC and CNF-LGO membrane-based LFEC. The inset is the magnified region at 120 Hz. **f** Schematic diagram of the coarse-grain model in kinetic Monte Carlo simulation. **g** Simulated ionic migration in TAS-LFEC and CNF-LGO membrane-based LFEC. The red arrow indicates the migration direction and displacement of ions.

remains the same, TAS-LFECs display $\varphi = -81°$ at 120 Hz, nearly identical to the commercial aluminum electrolytic capacitors, while for CS-LFECs, $\varphi$ drops to $-50°$ at 120 Hz, insufficient for efficient line filtering (Fig. 3c). Meanwhile, $C_A$ of TAS-LFECs is 4.8 mF cm$^{-2}$, 118% higher than that of CS-LFECs (2.2 mF cm$^{-2}$), and two orders of magnitude higher than that of aluminum electrolytic capacitors (Fig. 3d and Supplementary Fig. 17). In addition, control experiments are conducted adopting seven different commercial separators, TAS-LFECs demonstrate superior performance in both frequency characteristics and capacitance (Supplementary Fig. 18), achieving a high $C_A$ of up to 6.6 mF cm$^{-2}$ with $\varphi = -80°$ at 120 Hz (Fig. 3e and Supplementary Table 3)[1,12,18,19].

## Mechanisms underlying performance enhancement

Firstly, the thread-anchor structure of TAS shows significance in the separator integrity, which is the prerequisite of the normal function of LFECs (Fig. 4a). It is found that without GO anchors, the pristine CNF membrane-based LFECs have much higher leakage current than TAS-LFECs and quickly break down upon cycling (Supplementary Fig. 19). In contrast, TAS-LFECs maintain steady and show little attenuation of $\varphi$ and $C_A$ at 120 Hz after more than 1,000,000 cycles of galvanostatic charge-discharge (GCD) test (Supplementary Fig. 20). This can be attributed to the separator rupture-induced short circuit during assembly or cycling, given the separator thickness as low as 3 μm, close

to the size of bumps or impurities at electrode surfaces (Supplementary Fig. 21). SEM images of disassembled separators demonstrate cracks in the pristine CNF membrane, while TAS maintains surface integrity without cracks (Supplementary Fig. 22). To further unveil the in-solution stability of separators, we conducted an in-electrolyte AFM protrusion experiment (Fig. 4b). As shown in Fig. 4c, TAS can withstand a fold higher displacement against protrusion than the pristine CNF membrane. In electrolytes, the pristine CNF membrane cannot withstand swelling and separate away from each other, resulting in weak stress-resistant ability. GO anchors act as cross-linking points to bond CNFs altogether and reinforce the in-electrolyte stability, thus accounting for the enhanced stress-resistant ability of TAS.

In addition, the thread-anchor structure of TAS enables plenty of macropores as ionic transport paths, largely reducing the ionic resistance in the separator, $R_s$. We conducted a control experiment adopting large-sized (~10 μm) GO sheets, and found pores were blocked by the stack between CNFs and GO sheets (Supplementary Fig. 23), rendering severe hindrance for ionic transport and a fold increase in *SR* (Fig. 4d and Supplementary Fig. 24). As a result, the assembled device shows degradation of $\varphi$ at 120 Hz, from $-81°$ to $-69°$ (Fig. 4e). The kinetic behavior is also worsened, with the $b$ value changing from 0.96 to 0.89 (Supplementary Fig. 25). Kinetic Monte Carlo simulation visualizes the ionic kinetic behavior (Fig. 4f,

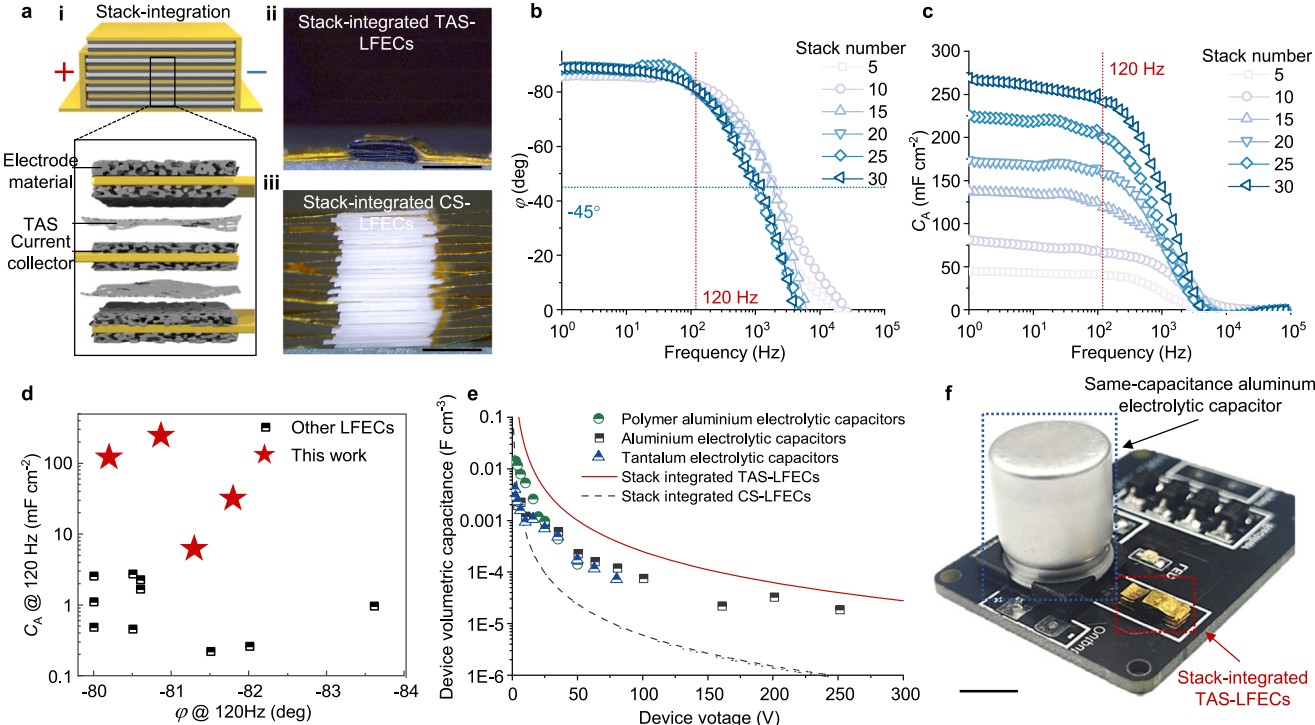

**Fig. 5 | Stack integration and high-power line filtering. a** Schematic diagram of stack-integrated TAS-LFECs (**i**). The enlarged image shows the detailed stack configuration. The optical images of a 30-layer stack-integrated TAS-LFEC (**ii**) and a 30-layer stack-integrated CS-LFEC (**iii**). Scale bar, 2 mm. **b** Bode diagrams of the stack-integrated TAS-LFECs with stack numbers varying from 5 to 30. **c** Plots of $C_A$ versus frequency of the stack-integrated TAS-LFECs with stack number varying from 5 to 30. **d** Comparison of $\varphi$ and $C_A$ at 120 Hz between stack-integrated TAS-LFECs and other LFECs reported previously. **e** Device volumetric capacitances of stack-integrated TAS-LFECs, stack-integrated CS-LFECs, and commercial electrolytic capacitors. **f** Optical images of stack-integrated TAS-LFECs (red dashed rectangle) and the same-capacitance commercial aluminum electrolytic capacitor (3.3 mF, blue dashed rectangle) Scale bar, 5 mm.

Supplementary Fig. 26, and Movie 1). In the same time frame, the ionic displacement in TAS is way larger than that in the CNF-LGO membrane, indicating a higher ionic migration rate and lower ionic resistance in TAS (Fig. 4g). Therefore, the performance improvement is well illustrated according to the equation

$$\varphi = -\arctan \frac{1}{\omega \cdot SR \cdot C} \tag{1}$$

At the specified frequency $\omega$ and capacitance $C$, $SR$ is reduced. Thus frequency characteristics are enhanced with phase angle $\varphi$ being lowered.

Furthermore, TAS thickness is another key. With the thickness descending from 15.0 μm to 3.5 μm, Nyquist curves continuously move from right to left, in tandem with the decrease of $SR$ at 120 Hz from 116 mΩ cm² to 36 mΩ cm² (Supplementary Figs. 27 and 28). In accordance, the frequency characteristic and capacitance are both improved, with $\varphi$ changing from −73° to −84° and $C_A$ increasing from 3.2 mF cm⁻² to 3.6 mF cm⁻² at 120 Hz. These results should be firstly attributed to the decline of ionic resistance in thin TAS, as verified by the EIS results of TAS (Supplementary Fig. 29). Meanwhile, finite elemental analysis reveals that the electric field strengthens as thickness decreases and promotes ionic migration (Supplementary Fig. 30). In addition, the decrease in separator thickness reduces the ionic resistance within electrode materials, which changes from 36 mΩ cm² to 11 mΩ cm². Kinetic Monte Carlo gives details of the built-in voltage, which indicates the voltage that is generated by the separated ions under the external voltage (Supplementary Fig. 31 and Movie 1). Note that the hysteresis of TAS-LFEC with 3.5 μm TAS is much lessened than that of 15 μm, indicating the boosted ionic migration rate resulting from the enhancement of the electric field[3]. In addition, a confirmatory

experiment is conducted by inserting two probes in TAS-LFECs to detect the built-in voltage, which well corroborates the simulated results (Supplementary Fig. 32).

### Stack integration of TAS-LFECs

As footprint is crucial for circuit miniaturization, significant efforts have been devoted to enhancing $C_A$[27]. However, the conventional strategy of thickening electrode materials inevitably compromises frequency characteristics[1,16,17]. Consistent with the previous reports, thickening electrode materials results in the increment of $SR$ at 120 Hz. Although $C_A$ improves from 5.5 mF cm⁻² to 9.4 mF cm⁻², $\varphi$ suffers a massive drop from −81° to −63° (Supplementary Figs. 33–35). Thanks to the low resistance, low thickness, and reliability of TAS, we exploit vertical space for parallel connection of units and achieve the stack integration of TAS-LFEC (Fig. 5a i). As shown in Fig. 5a ii, TAS saves 97% vertical space compared to CS (3 μm vs. 100 μm) in a single unit. Even with 30 units, the height of the stack-integrated TAS-LFECs is less than 1 mm. In contrast, while the stack integration for CS-LFECs can be conducted with stack number below 3, further stacking fails as the total thickness is too large to ensure the connection between current collectors (Fig. 5a iii).

Meanwhile, the stack-integrated TAS-LFECs behave in a good manner of parallel integration. As the stack number increases, $SR$ at 120 Hz decreases by a factor of stack number (Supplementary Fig. 36). More importantly, $C_A$ increases linearly without any observed degradation in frequency characteristics (Fig. 5b, c, and Supplementary Fig. 37), which is unachievable by solely thickening electrode materials. With $\varphi - −81°$, $C_A$ at 120 Hz reaches an utmost level of 240 mF cm⁻² (Fig. 5d), two orders of magnitude higher than the previous reports[1,12,18,19]. Considering the rating voltage of 1 V, the energy density of a 30-layer stack-integrated TAS-LFEC reaches 120 mJ cm⁻², even higher than stack-integrated dielectric capacitors[28]. By comparison, a

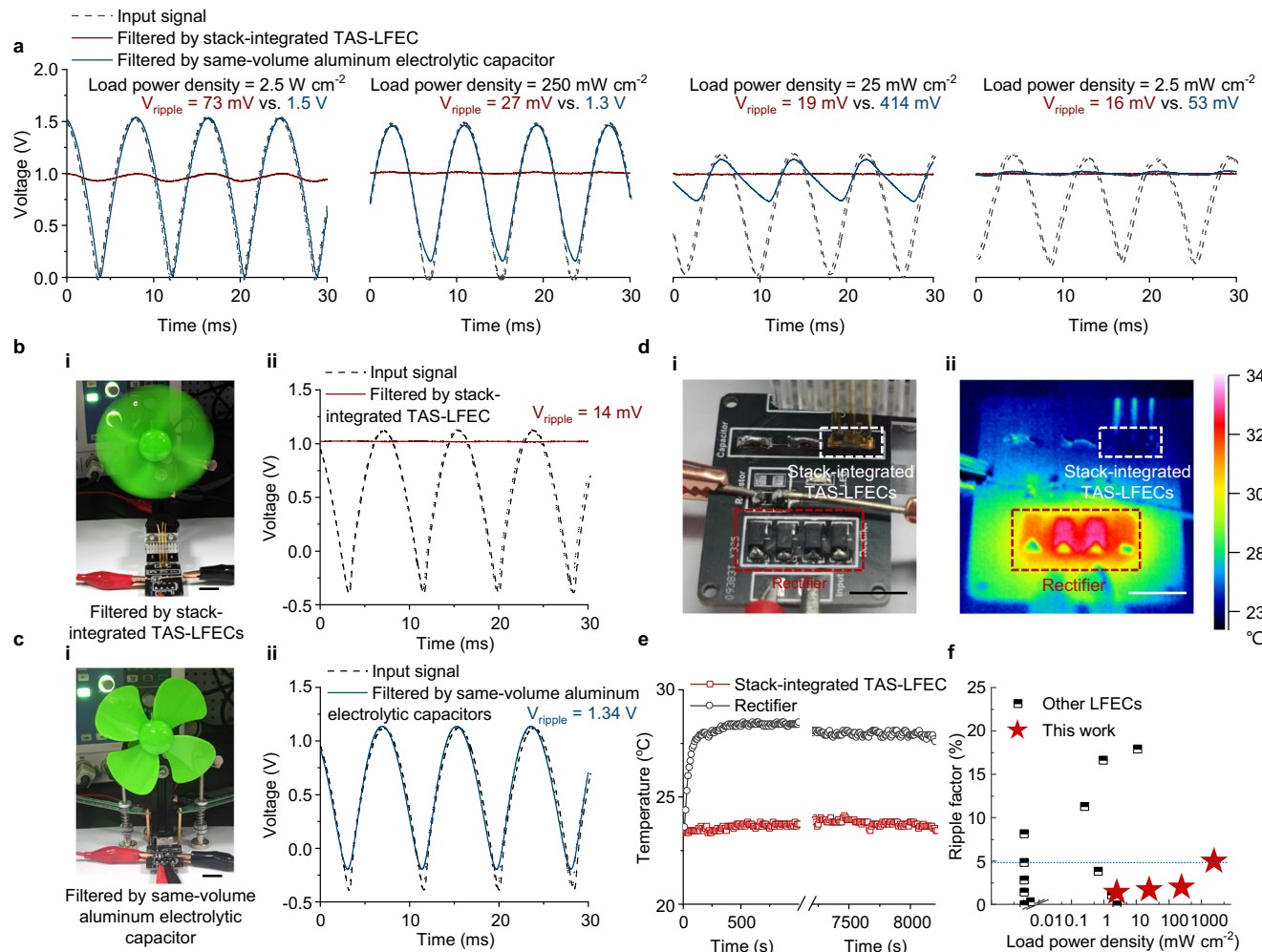

**Fig. 6 | High-power line filtering. a** The oscilloscope signals of input signals and output signals filtered by a 10-layer stack-integrated TAS-LFEC and the same-volume commercial aluminum electrolytic capacitor (10 μF) with the load power density varying from 2.5 mW cm$^{-2}$ to 2.5 W cm$^{-2}$. **b** Optical image (**i**) and oscilloscope signals (**ii**) of the on-load filtering test using a 20-layer stack-integrated TAS-LFEC. Scale bar, 1 cm. **c** Optical image (**i**) and oscilloscope signals (**ii**) of the on-load filtering test using the same-volume commercial aluminum electrolytic capacitor (10 μF). Scale bar, 1 cm. **d** Optical image (**i**) and infrared image (**ii**) of the

temperature tracing experiment of the rectifier-filter circuit with a load power density of 2.5 W cm$^{-2}$. The white and red dashed rectangles mark the 10-layer stack-integrated TAS-LFECs and the rectifier, respectively. Scale bar, 5 mm. **e** The surface temperature tracing of the stack-integrated TAS-LFECs and the rectifier with a load power density of 2.5 W cm$^{-2}$. **f** Comparison of load power density and ripple factor between stack-integrated TAS-LFECs and other LFECs reported previously. See details in Supplementary Table 1.

3-layer stack-integrated CS-LFEC shows $C_A$ of only 11 mF cm$^{-2}$ with $\varphi$ of −45° at 120 Hz, far from meeting the requirement of line filtering (Supplementary Fig. 38).

A practical comparison between LFECs and electrolytic capacitors at different rated voltages is the device's volumetric capacitance at a specific rated voltage. Because the capacitance decreases following $V^2$ for LFECs and $V$ for electrolytic capacitors, the stack-integrated TAS-LFECs take volumetric advantages over commercial electrolytic capacitors under 300 V (Fig. 5e). By contrast, the CS counterpart only takes advantage under 2 V due to the excessive occupation of separators. As shown in Fig. 5f, with the same capacitance, stack-integrated TAS-LFECs surpass aluminum electrolytic capacitor much in footprint and volume occupation, saving 10 times footprint and 160 times volume (12 mm³ vs. 1963 mm³).

### High-power line-filtering performance
The capability of LFECs to smooth no-load or low-power signals has been demonstrated in previous studies[8–13]. However, no examples can support practical high-power-consuming appliances such as central processing units, memory devices, direct current motors, etc. When

the load power density reaches 10 mW cm$^{-2}$ (Supplementary Table 1), ripple suppression becomes ineffective, resulting in a ripple factor larger than ~10%[8,12].

Benefiting from the low $SR$ and high $C_A$, stack-integrated TAS-LFECs put high-power line filtering into practice, and the performance is field-tested in a printed-circuit-board-grade rectifier-filter circuit, outperforming the aluminum electrolytic capacitor with the same volume, especially at high load power conditions (Fig. 6a). At a low-load condition (load power density of 2.5 mW cm$^{-2}$), little voltage ripple is observed both in output signals filtered by stack-integrated TAS-LFECs and the same-volume aluminum electrolytic capacitor (10 μF). When the load power density reaches 2.5 W cm$^{-2}$, the same-volume aluminum electrolytic capacitor loses control of ripple voltage, outputting the same signal as the input. Note that the ripple current density reaches 2 A cm$^{-2}$ (Supplementary Fig. 39), the 10-layer stack-integrated TAS-LFEC still suppresses the ripple factor under 5%, sufficing the requirement of effective line filtering.

In addition, we adopted an electric fan (corresponding to a load power density of 250 mW cm$^{-2}$) as the load to visualize the filtering performances. As shown in Fig. 6b and Movie 2, by employing a 20-

layer stack-integrated TAS-LFECs as the filter, the output signal shows a ripple voltage of only 14 mV, corresponding to a ripple factor of 1%. As expected, the electric fan works normally and smoothly. In contrast, when the commercial electrolytic capacitor with the same volume is used as the filter capacitor, the ripple voltage reaches 1.34 V, and the ripple factor comes to 90%, showing nearly the same signal as the input. In this case, the electric fan shakes a few times after the power supply and soon stops (Fig. 6c).

Furthermore, we traced the surface temperature of the stack-integrated TAS-LFECs. With a load power density of 2.5 W cm$^{-2}$, neither temperature increment nor performance degradation is observed for a 10-layer stack-integrated TAS-LFECs over more than 2 h (Fig. 6d and Supplementary Fig. 40), even though the rectifier raises 5 degrees higher (Fig. 6e). This can be attributed to the fact that the low $SR$ limits the alternate-current-induced heat generation, and the high stability of TAS prevents short circuits and side reactions, restricting the heat from direct currents. These results validate the practical application potential of the stack-integrated TAS-LFECs, especially for high-power scenarios. In comparison, taking a voltage ripple factor of 5% as the norm of effective line filtering, stack-integrated TAS-LFECs promote the tolerable load power density towards 2.5 W cm$^{-2}$, surpasses previous reports three orders of magnitude (Fig. 6f).

## Discussion

In this study, we proposed the significance of separators in line-filtering electrochemical capacitors. A highly ionic conductive separator is designed with a unique thread-anchor structure, which ensures both ionic transport and mechanical strength, realizing low resistance of down to 25 mΩ cm$^2$. By employing it in the device, $C_A$ reaches 6.6 mF cm$^{-2}$ with $\varphi = -80°$ at 120 Hz and could be boosted to 240 mF cm$^{-2}$ without frequency responses decay by TAS-enabled stack integration in parallel, overcoming the dilemma of thickening electrode materials. On this basis, high-power line filtering is achieved with reasonable ripple suppression. This work arouses a perspective on line-filtering electrochemical capacitors and promotes their practical applications for high-power scenarios.

## Methods

### Preparation of materials

**Preparation of GO nanosheets.** GO dispersion was synthesized through the modified Hummers' method[29] with high reproducibility. Typically, 5 g graphite powder (325 mesh, Qingdao Huatai lubricant sealing S&T Co., Ltd.) was mixed with 115 mL H$_2$SO$_4$ (98 wt%, AR, Sinopharm Chemical Reagent Co., Ltd) at 0 °C under mechanical stirring. Afterward, 15 g KMnO$_4$ (AR, Sinopharm Chemical Reagent Co., Ltd) was slowly added to the above-mixed solution, followed by 200 mL deionized water. Then, the mixture was transferred into a 95 °C oil bath and reacted for another 30 minutes. Next, the mixture was poured in 1,500 mL deionized water, and 30 wt% H$_2$O$_2$ (AR, Sinopharm Chemical Reagent Co., Ltd) solution was added subsequently to quench the reaction until no bubbles emerged. Before dialysis (cut off: 8000–14,000 Da, Viskase MD77, two weeks), the above mixture was filtered and washed with HCl solution (1 mol L$^{-1}$) 3 times. Finally, the purified GO suspension was centrifuged at 472 $g$ and 7552 $g$. For the consequent synthetic steps, GO sheets were torn apart into GO nanosheets by ultrasonication for 30 minutes using Branson Digital Sonifier SFX 250.

**Preparation of less defective GO for the electrode material.** Like the synthesis process of GO, less defective GO dispersion underwent a low-temperature procedure to reduce structural defects. For instance, 20 g deionized water was mixed with ice-cooled 230 mL H$_2$SO$_4$ (98 wt%) in a 500 mL three-necked flask at 0 °C under mechanical stirring, then adding 5 g graphite powder. Afterward, 15 g KMnO$_4$ was slowly added

to the above-mixed solution in 1 h. Then, the mixture reacted at 0 °C for another 48 hours. Afterward, the post-treatment procedures are the same as that of GO.

**Preparation of TAS.** Firstly, the dynamically fermented bacterial cellulose nanofibers (obtained from ScienceK) were diluted into 30 mL 0.17 mg mL$^{-1}$ solution. At agitated stirring, 0.1 mL 5 mg mL$^{-1}$ GO solution was added to the abovementioned solution. The mixture was then kept stirring for 30 min. Afterward, the solution was filtered by a Nylon membrane filter (obtained from Hangzhou Special Paper Industry Co., Ltd) to get the film. Time-lapse evaporation was conducted to slowly remove the solvent at 25 °C and high humidity of 80% ± 10% for 2 h. High-temperature vacuum drying was then performed in a vacuum drying oven at 60 °C with a vacuum degree below 0.2 kPa for 2 h to reinforce the interactions between CNFs and GO nanosheets. After that, TAS was obtained with high reproducibility.

**Fabrication of G/PEDOT film.** Firstly, the as-prepared less defective graphene oxide was diluted and mixed with LiClO$_4$ (AR, Aladdin) into a 2 mg mL$^{-1}$ GO and 0.1 M LiClO$_4$ solution as electrolyte. By using a large area Au plate as the counter electrode, and Ag/AgCl (CHI 111) as the reference electrode, reduced graphene oxide was deposited on the graphite foil at −1.1 V vs. Ag/AgCl. After being rinsed with deionized water to remove excessive GO, a deep reduction at −1.2 V vs. Ag/AgCl in 1 M LiClO$_4$ solution is conducted. After dialysis to remove residual ions, reduced graphene oxide was formed. After that, the hydrophobic surface of graphite foil changed to a super hydrophilic one. Secondly, when the surface of the reduced graphene oxide array remains wet but without excessive water, PEDOT:PSS solution (Clevios PH-1000) was drop cast onto it, forming a plat and uniform liquid layer. A uniform layer of G/PEDOT:PSS formed after drying. Next, G/PEODT film is obtained with high reproducibility by immersing G/PEDOT:PSS film in methanol solution for 1 minute and ensuing natural drying.

### Materials characterization

SEM images are obtained by Hitachi SU-8010 scanning electron microscope. TEM and HADDF-STEM images are obtained by FEI TecnaiTF20 transmission electron microscope. The electronic conductivity is tested by a four-point probe setup (GZKD KDY-1). X-ray photoelectron spectroscopy is conducted on ULVAC-PHI PHI Quantro SXM X-ray photoelectron spectrometer. The Raman spectrum is obtained from the HORIBA Evolution Raman spectrometer. Optical images are acquired on ZEISS Axio Scope. A1 microscope, RY605 microscope, and Canon EOS 80D digital single lens reflex.

### TAS-LFECs, stack-integrated TAS-LFECs and line-filtering circuits fabrication

**Construction of TAS-LFECs.** Firstly, the G/PEDOT film was cut by a laser (LAJAMIN LM-UVY15-3D) into individual electrodes of the desired size. Before the assembly, the as-prepared G/PEDOT films were immersed in the electrolyte (3 M H$_2$SO$_4$) for 10 minutes to infuse them with the electrolyte. The sandwich-type TAS-LFEC was assembled following the sequence: Au current collector, G/PEDOT, TAS, G/PEDOT, and Au current collector. The device reproducibility is good.

**Construction of stack-integrated TAS-LFECs.** Similar to the construction of TAS-LFECs, the stack integration can be realized by the stacking of components in the following sequence: Au current collector (left side), G/PEDOT, TAS, G/PEDOT, Au current collector (right side), G/PEDOT, TAS, G/PEDOT, Au current collector (left side), G/PEDOT, TAS, G/PEDOT, Au current collector (right side), ⋯. It is noted that the alignment and compression should be carefully done at every stacking procedure, and the short circuit of current collectors should be avoided, which is crucial for reproducibility.

**Construction of the PCB test circuit.** The PCB test circuit consists of an LED light bead, four diodes as a rectifier, and the resistance. The input is a sinusoidal signal (60 Hz) with variable peak-to-peak voltage ($V_{p-p}$) generated by an arbitrary function generator (Keysight 33500B). Typically, the resistance was $10\,\Omega$, and the $V_{p-p}$ of the rectified signal was 1.5 V. A 10-layer stack-integrated TAS-LFEC was used as the filter. The temperature distribution and variation were detected by the infrared imager (Fluke TiX640).

### Device characterization

**Electrochemical measurements.** Electrochemical performances are conducted at the electrochemical workstation (CHI 660e) based on the two-electrode test method. During the test, the environmental humidity is controlled around 50%, and the temperature is 25 °C. Cyclic Voltammetry (CV) is tested with scan rates varying from $1\,V\,s^{-1}$ to $2{,}000\,V\,s^{-1}$. GCD is tested at a current density ranging from $0.5\,mA\,cm^{-2}$ to $50\,mA\,cm^{-2}$. For the long-term stability test, a GCD current density of $100\,mA\,cm^{-2}$ is used, and the EIS test is performed at the interval of every 1000 cycles.

**EIS measurements.** The EIS test is conducted by an electrochemical workstation (CHI 660e and AMETEK PARSTAT 3000 A). The initial voltage is 0 V, amplitude is 5 mV, and frequency range varies from 10,000 Hz to 1 Hz, with 12 testing points per decade frequencies. It is noted that the testing wires could introduce excessive resistance into the measurement, which should be subtracted by conducting a blank control.

**Oscillogram measurements.** Oscillograms are recorded by ROH-DE&SCHWARZ RTB2002, DC mode, with a sampling rate of 83.3 KSa ms$^{-1}$.

**Electrochemical performance calculations.** Based on the data from electrochemical and EIS measurements, the performance parameters are derived as follows:

The real part and imaginary part of capacitance is calculated by,

$$C_{\text{Re}}(f) = \frac{-Z_{\text{Im}}(f)}{2\pi M f |Z(f)|^2} \tag{2}$$

$$C_{\text{Im}}(f) = \frac{Z_{\text{Re}}(f)}{2\pi M f |Z(f)|^2} \tag{3}$$

Where $C_{\text{Re}}(f)$ and $C_{\text{Im}}(f)$ are the real part and imaginary part of capacitance at a specified frequency $f$, $Z_{\text{Re}}(f)$ and $Z_{\text{Im}}(f)$ are the real part and imaginary part of impedance $Z(f)$ at a specified frequency $f$, which is typically 120 Hz. $M$ is the area or volume of LFECs. For areal capacitance $C_A(f) = C_{\text{Re}}(f)$, $M$ refers to the device area, typically $2 \times 2$ mm; for volumetric capacitance $C_V(f) = C_{\text{Re}}(f)$, $M$ refers to all the volume occupied, including electrodes, separators, and current collectors.

### Theoretical calculation

**Finite elemental analysis.** We adopted the software COMSOL to simulate the external electric field in LFECs. Physical fields of electrostatics and electric currents were used. Conductivity is $500\,S\,cm^{-1}$ for G/PEDOT and $2\,S\,m^{-1}$ for the electrolyte. The dielectric constant is 80 for G/PEDOT and 78 for the electrolyte[30–32]. The model was established as entirely identical to the realistic TAS-LFECs. For the boundary conditions, according to the experiments, the voltage of one electrode was assigned to 0 V; the opposite, 1 V. We calculated the cases of separator thickness varying from $3.5\,\mu m$ to $15\,\mu m$, and the corresponding areal resistances were calculated based on Ohm's law.

**Kinetic Monte Carlo simulation.** N-Fold Way (NFW) algorithm is applied to perform kinetic Monte Carlo simulations. The details about this kinetic Monte Carlo and NFW algorithm could be found in references[33–35]. The rate of a diffusional hop is

$$r_D = \begin{cases} D_0 & \Delta E \leq 0 \\ D_0 \exp\left(-\frac{\Delta E}{k_B T}\right) & \Delta E > 0 \end{cases} \tag{4}$$

where $\triangle E$ represents energy change induced by hop. The total energy of our coarse grain model is given by

$$E_{tot} = \alpha \sum_{i,j=ions} \frac{Z_i Z_j}{r_{ij} + d_c} + \sum_j Z_j V_{ext} + E_{ad} \sum_k n_k \tag{5}$$

where the first term is used to describe the interaction between ions, and $\alpha$ is a scaling factor. Its only difference from coulomb potential comes from $d_c$, which is set to guarantee that the coulomb attraction at the time of ions' collision is small enough to be broken by kinetic energy ($k_B T$). Note that this is a coarse grain model, the values of $d_c$ and $\alpha$ used in our simulation are set to $1\,\mu m$ and $0.01\,eV\cdot\mu m$. The second term represents the energy of ions under the external electric field, where $V\_ext$ is the electric potential given by two electrodes with uniform charge density ($2.88 \times 10^{-6}\,e\,\mu m^{-3}$). Besides, the networks containing PEDOT and GO are considered as rigid skeletons, ions could be adsorbed if the distance between ions and PEDOT sites is no more than $1.2\,\mu m$, and adsorption energy $E_{ad}$ is set to $-0.18$ eV.

For the impedance calculation, the built-in voltage could be monitored in kinetic Monte Carlo simulations by calculating the electric potential difference for each step. It is proportional to the charge number accumulating at the electrodes, and thus its partial derivative with respect to x could reflect the dynamic evolution of electric current. The impedance could be obtained according to its definition, i.e. $Z = V/I$, where $V$ and $I$ are the voltage and electric current, and both of them are complex-valued functions.

## Data availability

Source data are provided with this paper. Additional data related to this work are available from the corresponding authors upon request. Source data are provided with this paper.

## Code availability

The code supporting this study is available from the corresponding author upon request.

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

## Acknowledgements

This work was supported by the National Key R&D Program of China (2024YFB4609100, 2024YFF0506000 to L.Q.), National Natural Science Foundation of China (22035005 to L.Q., 223B2903 to Y.H., 52090032 to H.C., 11972349 to F.L., 11790292 to F.L., 22105040 to M.W.), Strategic Priority Research Program of the Chinese Academy of Sciences (XDB22040503 to F.L.), Fujian Science & Technology Innovation Laboratory for Optoelectronic Information of China (2021ZZ127 to M.W.), Natural Science Foundation of Fujian Province of China (2021J01588 to M.W.).

## Author contributions

Conceptualization: L.Q. and Y.H.; Methodology: Y.H.; Investigation: Y.H., P.L., G.L., B.L., H.W., H.C., M.W., F.L. and Z.D.; Visualization: Y.H.; Funding

acquisition: L.Q., H.C., M.W., F.L. and Y.H.; Project administration: L.Q.; Supervision: L.Q.; Writing original draft: Y.H., F.L. and L.Q.

## Competing interests

The authors declare no competing interests.
