## [Transparent Peer Review file · Nature Communications]

Separator with high ionic conductivity enables electrochemical capacitors to line-filter at high power

Corresponding Author: Professor Liangti Qu

Version 0:

Reviewer comments:

Reviewer #1

(Remarks to the Author)

This work identifies the commonly used thick separator as a bottleneck for developing Line-filtering electrochemical capacitors (LFECs) since this thick separator contributes a large ionic resistance. By identifying this problem, the authors developed ultrathin (3 μm thick) separators with high ionic conductivity, enabling LFECs with a large area capacitance density of 6.6 mF/cm^2 and a phase angle of -80° . This is a breakthrough for LFECs. The following comments are provided for the authors to consider and address:

1. A major concern is the reliability and leakage of the ultrathin separator. The SEM images show pore sizes on the 1 μm scale, and SI Fig. 18 shows a continuous decrease in voltage, starting at 0.8 V. Please provide more complete information on this ultrathin separator. Does the failure of the device occur due to the electrode or the separator after 500k cycles? Does the separator improve the long-term cycling stability of the device?
2. Does the defective graphene in the separator limit the maximum voltage to 0.8 V?
3. The term "stack integration" is misleading, particularly since the abstract emphasizes, "TAS-enabled stack integration breaks the trade-off between capacitance and frequency response, boosting the areal capacitance to 240 mF cm^{-2} without decay of frequency characteristics." It is suggested to clarify this as "stack integration in parallel."
4. For Fig. 3b fitting, the CV plots should be presented.
5. EIS in SI Fig. 33 appears quite different from Fig. 3a. Please provide more detailed information on the electrode, particularly its thickness. Can the nanoscale features in the electrode penetrate the separator?
6. In Fig. 5e, the equation and how it was derived should be explained.

Reviewer #2

(Remarks to the Author)

Hu et al. report a highly ion-conductive thread-anchor structured separator (TAS) based on the interactions between cellulose nanofibers (CNFs) and graphene oxide (GO) nanosheets. The thread-anchor architecture combines abundant macropores with strong mechanical properties. By conducting TAS-enabled 3-dimensional stack-type integration, CA is improved largely without compromising frequency responses, effectively solving the dilemma of making thicker electrode materials. The manuscript is well structured, and the results well analyzed. However, the followed points need to be addressed before this paper can be accepted:

1. The authors said that "In contrast, while the stack integration for CS-LFECs can be conducted with stack number below 3, further stacking fails as the total thickness is too large to ensure the connection between current collectors (Fig. 5a iii)." What is the Bode diagrams of the stack-integrated CS-LFECs with stack numbers varying from 1-3 (Figure 5b) and plots of CA versus frequency of the stack-integrated CS-LFECs with stack number s varying from 1-3 (Figure 5c)? These control experiments should be added to better explain the performance of the stack TSA-LFECs.
2. The authors said "With a load power density of 2.5 W/cm^2 , neither temperature increment nor performance degradation is observed for a 10-layer stack-integrated TAS-LFECs over 365 more than 2 hours (Fig. 5j and Supplementary Fig. 40)" (line

- 364). What happens if higher power density is applied? What is needed for use in practical applications?
3. For Figure 2 c-e, scale bars should be added into the images.
4. The author tested the stress-strain curves and Nyquist diagram of pristine CNF, commercial cellulose separator (NKK-TF4030), and TAS. Pristine GO should also be added as a control experiment.
5. For Figure 3b, what is R_2 of the calculated b value for both TAS-LFEC and CS-LFEC? What happens to CS-LFEC with a high scan rate?

Reviewer #3

(Remarks to the Author)

Dear Authors:

The manuscript by Hu et al has presented a new separator that allows robust operation at low film thickness and thereby improves the frequency response of electrochemical capacitors. Majority of data and analysis support their claims well. There are some points below to clarify:

1. We agree that reducing separator thickness is important, the area of the capacitors will also reduce series resistance (resistors in parallel) in a cell. For example, in commercial devices with larger areas, the ESR naturally decreases due to the larger surface area. More discussion is needed to explain the effect of thin separators when using for large area devices. For fair comparison, it is crucial to provide the device area (before normalization) and compares to the total electrode area in AEC.
2. Analyzing self-discharge performance is crucial for this manuscript due to the ultra-thin 3 μm separator. In Figure S18, is y-axis unit really A cm^{-2} because that is quite high. Additionally, performing a Coulombic efficiency analysis at different current densities is necessary to better understand potential energy losses associated with the thin separator. A comparison of leakage current and self-discharge performance with a commercial separator is also needed.
3. In this paper, the ESR of the devices is below 0.05 ohms, approaching the electrical resistance of the conductive wires in the characterization equipment. It's important to describe the method used to mitigate the influence of resistance from the conductive wires. Is that the reason why Au current collectors were used? But Au is impractical for commercial applications, so it would be useful to explain the rationale of using Au.
4. In Fig 5g, we need more explanation on the comparison between Al electrolytic capacitor (AEC) and the capacitor (TAS-LFEC) in this work. The authors mentioned that both capacitors have the same capacitance (also please add that specific value in the caption), and the AEC and TAS-LFEC had the same phase angle in Fig 3c. Then RC constant should remain the same, since load was 10 Ohm. Please elaborate on the reason the AEC fail to rectify if the phase angle and RC constant are the same as TAS-LFEC?
5. Please comment on the separator uniformity (edge vs center). Has the separator been tested with electrolytes with alkaline pH?
6. Did the authors evaluate separator performance with electrodes with thick/high loading, with a loading range similar to those of commercial supercapacitors? At what electrode thickness/loading will the separator effect become negligible as resistance is dominated by electrodes?
7. As a future design rule, please comment on the design rule for the separator, such as the ratio of GO to cellulose content to optimize the pore size. What is the optimal GO size so that it can acts an anchor while keeping pore size large enough for ion transport.

Editing issues:

8. In Fig 1b, the box plot was described in the text that the separator accounts for 60% of the total series resistance. This message is important and it can be observed that the ionic resistance of electrodes and separators are similar. However, how was this exact percentage calculated, when the data were quite spread out? Was it from comparing the median lines?
9. The X and Y axes in all EIS results within this manuscript use different scales, which should be corrected for consistency. The EIS results of TAS and CNF-LGO in Figure 4d display distinct behaviors at high frequencies; further explanation is needed to clarify these differences and how they relate to the separator design. In Figure 5b and 5c, please explain the appearance of bumps in the 10-100 Hz region.
10. In Figure 5d, please indicate how many layers were used in the cell structures for all the cited papers.
11. In Figure S19, please specify how long the cycling test took to complete. Additional details, including the device's capacitance, GCD curves before and after cycling, and EIS results, are necessary to support the cycling stability conclusions.
12. In Table S2, averaging data from different papers has no clear physical meaning and should be reconsidered.
13. The fonts in all the plots are too small and not legible. Please increase the font size to make them readable without the need to magnify.

14. The grammar in the title needs correction. Please edit “ultra-highly” and “high-load filter”. Perhaps the authors mean “Separator with ultrahigh ionic conductivity in electrochemical supercapacitors for ac line filtering (at high power)”. There are other instances where the text needs to improve; please check. And perhaps add in the introduction that “line-filtering” means alternating current (ac) line filtering, even though it might be obvious to us in the field, it is still better to clarify it as ac line filtering.

Reviewer #4

(Remarks to the Author)

Version 1:

Reviewer comments:

Reviewer #1

(Remarks to the Author)

The authors have adequately addressed my previous comments, and the manuscript is now suitable for acceptance. Below are a few minor comments for further improvement:

Line 332 and 334: The unit $F_{cm}(-3)V(2)$ is used for capacitance density, which is scientifically incorrect. While it is understood that achieving high rated voltage requires connecting multiple devices in series—resulting in a volumetric capacitance decrease proportional to the square of the voltage—such a unit is not appropriate.

Figure 5g: The TAS-EC device is reported to have dimensions of $2\text{mm} \times 2\text{mm} \times 26.4\ \mu\text{m} \times 10$. Could the authors confirm whether the AEC device has similar dimensions? Additionally, it is noted that the maximum applied voltage for TAS-EC is 1.5 V, not the rated 1 V as previously indicated.

"Built-in Voltage": The term "built-in voltage" has been used in a manner that differs from the conventional definition (e.g., in semiconductor physics). Can the authors clarify its specific meaning within the context of this manuscript?

Reviewer #2

(Remarks to the Author)

The revisions make this manuscript suitable for publication in Nature Communications.

Reviewer #3

(Remarks to the Author)

This revision has satisfactorily addressed the reviewers' comments.

Reviewer #4

(Remarks to the Author)

Reviewer #1 (Remarks to the Author):

This work identifies the commonly used thick separator as a bottleneck for developing Line-filtering electrochemical capacitors (LFECs) since this thick separator contributes a large ionic resistance. By identifying this problem, the authors developed ultrathin (3 μm thick) separators with high ionic conductivity, enabling LFECs with a large area capacitance density of $6.6 \text{ mF}/\text{cm}^2$ and a phase angle of -80° . This is a breakthrough for LFECs. The following comments are provided for the authors to consider and address:

Reply:

Thank you for recognizing our research and providing valuable suggestions. We have revised our Manuscript based on these suggestions. Please check the revised sections highlighted in red font for your consideration. Below are our point-by-point responses to your comments.

1. A major concern is the reliability and leakage of the ultrathin separator. The SEM images show pore sizes on the $1 \mu\text{m}$ scale, and SI Fig. 18 shows a continuous decrease in voltage, starting at 0.8 V . Please provide more complete information on this ultrathin separator. Does the failure of the device occur due to the electrode or the separator after 500k cycles? Does the separator improve the long-term cycling stability of the device?

Reply:

Thanks for your kind suggestion. More detailed information has been added to the revised Manuscript and Supplementary Information file.

(1) Please provide more complete information on this ultrathin separator.

To clearly show the detailed thread-anchor structure (Fig. R1 a), we have to decrease the thickness of

TAS, in which the pores seem so large that puncture can easily happen. We noted this in the caption in the revised manuscript. The top-view SEM image of the normal-thickness TAS membrane is Fig. R1 b. This is the result of multiple stack structures of the thread-anchor structures, which can be effective in resisting puncture. In detail, we added an experiment to compare the leakage current and self-discharge behaviors between the pristine CNF membrane, commercial separator, and TAS. As shown in Fig. R1 c-e, TAS has a lower leakage current than the pristine CNF membrane and behaves similarly to the commercial separator. Although the leakage is unavoidable, neither leakage current nor self-discharge has a noticeable effect on the line-filtering performance. This is because the magnitude of the leakage current ($\sim 10^{-5}$ A cm⁻²) is way lower than the working-mode current (~ 0.1 A cm⁻²) (Fig. R1 c); and in the charge and discharge cycles, self-discharge has nearly no impact (Fig. R1 e).

(2) Does the failure of the device occur due to the electrode or the separator after 500k cycles?

TAS-LFEC didn't fail after 500k cycles. In a repeated experiment, it can sustain more than one million cycles (Fig. R1 f). The reason why more cycles are not being conducted is the limitation of the encapsulation method. The evaporation of electrolytes needs to be taken into consideration and should be addressed through technical engineering optimization in the following research.

The pristine CNF membrane-based LFEC failed after around 100k cycles. The reason is the puncture of the separator. As shown in Fig. R1 g and h, the phase angle at 120 Hz shows a noticeable decrease, and the leakage current runs up the working-mode current level, rendering the failure of charge.

(3) Does the separator improve the long-term cycling stability of the device?

In conclusion, TAS improved the long-term cycling stability compared to the pristine CNF membrane-based LFEC.

Fig. R1 | **a,b**, SEM images of TAS with a lesser thickness (**a**) and normal thickness (**b**). **c-e**, Comparison of leakage current density (**c**) and self-discharge behavior (**d,e**) between Pristine CNF membrane, Commercial separator, and TAS. **f,g**, Cyclic tests between TAS-LFEC (**f**) and pristine CNF membrane-based LFEC (**g**). **h**, Galvanostatic curves of the Pristine CNF membrane-based EC before and after breaking.

2. Does the defective graphene in the separator limit the maximum voltage to 0.8 V?

Reply:

No, the maximum voltage is mainly dominated by the electrochemical stable voltage window of the electrolyte (3 M H₂SO₄ solution here). The defects in graphene oxide are essential to making them electronically insulated, which ensures the devices' normal operation without internal short circuits.

3. The term "stack integration" is misleading, particularly since the abstract emphasizes, "TAS-enabled stack integration breaks the trade-off between capacitance and frequency response, boosting the areal capacitance to 240 mF cm^{-2} without decay of frequency characteristics." It is suggested to clarify this as "stack integration in parallel."

Reply:

Thank you for the kind suggestion. We have changed the terminology in the revised Manuscript.

4. For Fig. 3b fitting, the CV plots should be presented.

Reply:

We fitted Fig. 3b by applying cyclic voltammetry on TAS-LFEC and CS-LFEC with the scan rate varying from 1 V s^{-1} to $1,000 \text{ V s}^{-1}$, and by reading the cathodic and anodic current values at 0.4 V . Please check Fig. R2 for details. We have also added those figures to the revised Supplementary Information file.

Fig. R2 | Cyclic voltammetry plots of the TAS-LFEC (a) and CS-LFEC (b) at varying scan rates from 1 V s^{-1} to $1,000 \text{ V s}^{-1}$.

5. EIS in SI Fig. 33 appears quite different from Fig. 3a. Please provide more detailed information on the electrode, particularly its thickness. Can the nanoscale features in the electrode penetrate the

separator?

Reply:

Supplementary Fig. 33 and Fig. 3a seem different because of the different axes' scales. Actually, Nyquist diagrams should be presented with axes' scales being kept the same. However, to clearly show the evolution of performances of all the samples in a single image, we have to rescale Supplementary Fig. 33.

If we adjust Fig. 3a into the same frame as Supplementary Fig. 33 (Fig. R3 a,b), Their curves are similar. The origin of the twist should be ascribed to the cascade ionic migration within the electrode materials, rendering the overlap of multiple impedance arcs. This information is negligible when the X-axis is not magnified (Fig. R3 c,d).

Fig. R3 | Comparison between Fig. 3a and Supplementary Fig. 33 at the same axes' scales. **a**, Nyquist diagram of TAS-LFEC and CS-LFEC. **b**, Nyquist diagram of TAS-ECs with varying electrode thicknesses. **c**, High-frequency region of Nyquist diagram of TAS-LFEC and CS-LFEC. **d**, High-frequency region of Nyquist diagram of TAS-ECs with varying electrode thicknesses.

As for the characterization of the electrode, please see Fig. R4 a for the cross-sectional details. The thickness varies from 3 to 8 μm according to the liquid amount of drop casting. More detailed

information is given by the 3-dimensional white light interferometry (Fig. R4 b-d). We construct a blank line for thickness detection using laser scribing, which could burn out the electrode materials but have no harm to the substrate (silicon wafer). As shown from the height distribution, a typical electrode in this work has a thickness of 4.2 micrometers and surface bumps of about 500 nanometers. These bumps originate from the vertical graphene within the electrodes or the impurities on the electrode surfaces.

Fig. R4 | Morphology characterization of the electrode. **a**, Cross-sectional SEM image of the electrode. **b**, Top-view optical image of the electrode for white light interferometry 3-dimensional morphology test. **c**, Height distribution image of the electrode. **d**, The height distribution diagram of the detect line in **c**.

After cycling, the electrode may penetrate the separator. The separator is 3-micrometer thick, and the nanoscale features of electrodes are 500 nanometers. Direct penetration is less possible since the

bumps/spikes will not grow during the cycling test. However, when the assembly stress is applied to the device, the electrodes' expansion and contraction during cycling could lead to non-uniform extrusion and, thus, penetration.

We observed penetration in the pristine CNF membrane after cycling (Fig. R5 a). Some still-linked CNF fibers near the hole indicate that they are torn apart. For comparison, we also examined the TAS membrane after the same cycling test (Fig. R5 b). Non-uniform extrusion can also be observed, but penetration is not observed, indicating the strength advantages of the thread-anchor structure compared to the pristine CNF membrane.

Fig. R5 | Top-view SEM images of the separators after the test. **a**, The pristine CNF membrane. The red dashed circle indicates the protrusion induced by the assembly stress. **b**, The TAS.

6. In Fig. 5e, the equation and how it was derived should be explained.

Reply: We first apologize for a clerical mistake. The unit of the device's volumetric capacitance should be $F\text{ cm}^{-3}$. We have corrected this in the revised Manuscript.

The following is the detailed calculation process (which is also appended in the revised Supplementary Information):

The real part of capacitance is calculated by,

$$C_{\text{Re}}(f) = \frac{-Z_{\text{Im}}(f)}{2\pi Mf|Z(f)|^2},$$

Where $C_{Re}(f)$ is the real part of capacitance at a specified frequency f , $Z_{Re}(f)$ is the real part of impedance $Z(f)$ at a specified frequency f , f is typically 120 Hz, and M is the normalization factor. For the device volumetric capacitance, $C_V(f)$. $C_V(f) = C_{Re}(f)$, when M equals all the volume occupied, including electrodes, separators, current collectors.

The device volumetric capacitances at a specific voltage rating V are calculated by,

$$C_V(f, V) = C_V(f) \times v_d^2 / V^2,$$

where $C_V(f, V)$ is the volumetric capacitance at a specific voltage rating, $C_V(f)$ is the volumetric capacitance of a single unit, v_d is the voltage of a single unit, and V is the voltage rating of the integrated device.

For example, for a TAS-EC, the real part capacitance at 120 Hz, $C_{Re}(120 \text{ Hz})$, is 265 μF , device area is $0.2 \text{ cm} \times 0.2 \text{ cm}$, device thickness comprising a separator (3 μm), two electrodes ($4.7 \mu\text{m} \times 2$), two current collectors ($7 \mu\text{m} \times 2$) is 26.4 μm . Therefore, the device volumetric capacitances at 120 Hz, $C_V(120 \text{ Hz})$, is 2.5 F cm^{-3} . Given the device voltage of 1 V, the device volumetric capacitances at 120 Hz at a voltage rating, $C_V(120 \text{ Hz}, V)$, is $2.5 \text{ F cm}^{-3} \times 1\text{V} \times 1\text{V} = 2.5 \text{ F cm}^{-3} \text{ V}^2$. If the integrated TAS-ECs have a voltage rating of 10 V, then the volumetric capacitance should be $2.5 \text{ F cm}^{-3} \text{ V}^2 / 10 \text{ V} / 10 \text{ V}$, equaling 0.025 F cm^{-3} .

In a similar way, for a commercial electrolytic capacitor having a cylinder shape, with a diameter of 6.3 mm, a height of 8 mm, a voltage rating of 2 V, and a capacitance of 1000 μF , the device volumetric capacitance $C_V(120 \text{ Hz})$ should be $1000 \mu\text{F} / (1/4 \times \pi \times 6.3 \text{ mm} \times 6.3 \text{ mm} \times 8 \text{ mm}) = 0.004 \text{ F cm}^{-3}$.

Reviewer #2 (Remarks to the Author):

Hu et al. report a highly ion-conductive thread-anchor structured separator (TAS) based on the interactions between cellulose nanofibers (CNFs) and graphene oxide (GO) nanosheets. The thread-anchor architecture combines abundant macropores with strong mechanical properties. By conducting TAS-enabled 3-dimensional stack-type integration, CA is improved largely without compromising frequency responses, effectively solving the dilemma of making thicker electrode materials. The manuscript is well structured, and the results well analyzed. However, the followed points need to be addressed before this paper can be accepted:

Reply:

Thank you for your positive comments on our Manuscript and your suggestions to help us further improve it. We have revised our Manuscript accordingly and please check the revised parts in red font.

The following are the point-to-point replies to the comments.

1. *The authors said that “In contrast, while the stack integration for CS-LFECs can be conducted with stack number below 3, further stacking fails as the total thickness is too large to ensure the connection between current collectors (Fig. 5a iii).” What is the Bode diagrams of the stack-integrated CS-LFECs with stack numbers varying from 1-3 (Figure 5b) and plots of CA versus frequency of the stack-integrated CS-LFECs with stack number svarying from 1-3 (Figure 5c)? These control experiments should be added to better explain the performance of the stack TSA-LFECs.*

Reply:

Thank you for your kind suggestion. We have added the corresponding figures in the revised Supplementary Information file. As shown in Fig. R6 a,b, the stack-integrated CS-LFECs have the

same frequency characteristics as the single-unit CS-LFEC. The phase angle is around -45° , far from the requirement of high-efficiency filtering. Meanwhile, the areal capacitances of the stack-integrated CS-LFECs at 120 Hz are also much lower than those of TAS-LFECs. With the stack number of 3, the stack-integrated CS-LFEC has only 11.4 mF cm^{-2} , much lower than that of the stack-integrated TAS-LFEC (23.7 mF cm^{-2}), even though they have similar capacitances at low frequency, which means the electrode materials are the same.

Fig. R6 | Comparison of electrochemical performances between the stack-integrated TAS-LFECs and CS-LFECs with stack numbers varying from 1-3. **a**, Bode diagrams of the stack-integrated CS-LFECs with stack numbers varying from 1-3. **b**, Plots of areal capacitance versus frequency of the stack-integrated CS-LFECs with stack numbers varying from 1-3. **c**, Bode diagrams of the stack-integrated TAS-LFECs with stack numbers varying from 1-3. **d**, Plots of areal capacitance versus frequency of the stack-integrated TAS-LFECs with stack numbers varying from 1-3.

2. The authors said “With a load power density of 2.5 W/cm^2 , neither temperature increment nor performance degradation is observed for a 10-layer stack-integrated TAS-LFECs over 365 more than 2 hours (Fig. 5j and Supplementary Fig. 40)” (line 364). What happens if higher power density is

applied? What is needed for use in practical applications?

Reply:

There are two consequences of increasing the load power density: 1) ripple voltage gets more extensive, and 2) heat generation increases.

The practical application mainly concerns the issues mentioned above. The amount of load power depends on the electronic circuit design, but it is always desired to use less space for filtering capacitors in a circuit.

Here, we used the rectifier-filter circuit for filtering performance measurement. By decreasing the resistance from 10,000 Ω to 10 Ω , the load power density increases accordingly. Our testing equipment limits higher load power density because the rectifier's inner resistance causes a severe voltage drop with a load resistance of 10 Ω , and 1 V voltage output cannot be achieved if the load resistance is lowered. In fact, the current load power density we used is comparatively high, enough to activate an electric motor that requires high initial power. As shown in Fig. R7, both the ripple voltage and the heat generation issues are well controlled by the stack-integrated TAS-LFEC.

In conclusion, the stack-integrated TAS-LFEC can afford line-filtering work with a high load power density for practical applications.

Fig. R7 | **a**, The oscilloscope signals of input signals and output signals filtered by a 20-layer stack-integrated TAS-LFEC and the same-volume commercial aluminum electrolytic capacitor with the load power density varying from 2.5 mW cm^{-2} to 2.5 W cm^{-2} . **b**, Optical image (i) and infrared image (ii) of the temperature tracing experiment of the rectifier-filter circuit with a load power density of 2.5 W cm^{-2} . The white and red dashed rectangles mark the 10-layer stack-integrated TAS-LFECs and the rectifier, respectively. Scale bar, 5 mm. **c**, The surface temperature tracing of the stack-integrated TAS-LFECs and the rectifier with a load power density of 2.5 W cm^{-2} .

3. For Figure 2 c-e, scale bars should be added into the images.

Reply: We have added scale bars to the images in the revised Manuscript.

4. The author tested the stress-strain curves and Nyquist diagram of pristine CNF, commercial cellulose separator (NKK-TF4030), and TAS. Pristine GO should also be added as a control experiment.

Reply:

Thank you for your suggestion. The stress-strain curves and Nyquist diagrams of the pristine GO membrane are added as the control experiment in the revised Supplementary Information.

Because GO is an additive and takes little proportion in the TAS separator, the pristine GO membrane has distinct differences from the TAS separator. The only existence of the GO sheets leads to a layer-by-layer assembly, leaving the membrane with poor mechanical properties (tensile strength of 37 MPa) and ionic conductivity (ionic resistance of $169 \text{ m}\Omega \text{ cm}^2$) (Fig. R8).

Fig. R8 | **a**, Stress-strain curve of pristine the GO membrane. **b**, Nyquist diagram of pristine the GO membrane in the electrolyte of 3M H_2SO_4 . **c**, Stress-strain curves of commercial cellulose separator (NKK-TF4030), pristine GO, pristine CNF, and TAS. **d**, Nyquist diagram of commercial cellulose separator, pristine GO, pristine CNF, and TAS in the electrolyte of 3M H_2SO_4 .

5. For Figure 3b, what is R^2 of the calculated b value for both TAS-LFEC and CS-LFEC? What happens to CS-LFEC with a high scan rate?

Reply: The square R of the regression for TAS-LFEC and CS-LFEC are 0.9989 and 0.9343 (Fig. R9).

Fig. R9 | Plot of the logarithm of current density (i) versus the logarithm of scan rates (v) for CS-LFEC (a) and TAS-LFEC (b). The red dashed line is the fitting curve.

Regarding the second question, perhaps the reviewer refers to the current decrease at a high scan rate. This wouldn't happen in a normal situation. But in this case, with a high scan rate, CS-LFEC has severe polarization in the cyclic voltammetry curves. The reason for this is the overlarge ionic resistance of the CS separator. The induced capacitance loss renders the decrease in current with the increase in scan rate. (Fig. R10).

We conducted a calculation for clarity. The LFECs here can be simplified as a resistor-capacitor component with a resistance of R and a capacitance of C . If we apply cyclic voltammetry with a scan rate of v on the capacitor, the relationship between the voltage u and the current i can be deduced: $i = vC(1 - e^{-\frac{u}{RCv}})$. In normal situations, C nearly remains the same. Therefore, the current i should increase monotonously with the increase in v (Fig. R10). However, the overlarge capacitance of CS separator leads to a severe capacitance loss (we anticipate the loss from 6.6 mF cm^{-2} to 3 mF cm^{-2} here). Consequently, the current at 1000 V s^{-1} becomes lower than that at 500 V s^{-1} (Fig. R10 b).

Fig. R10 | **a**, Cyclic voltammetry plots of the CS-LFEC at varying scan rates from 1 V s^{-1} to $1,000 \text{ V s}^{-1}$. The red dash indicates the voltage where b value comes from **b**, The calculated cyclic voltammetry with R of $100 \text{ m}\Omega$, C of 6.6 mF cm^{-2} or 3 mF cm^{-2} , and ν varying from 100 V s^{-1} to $1,000 \text{ V s}^{-1}$.

Reviewer #3 (Remarks to the Author):

Dear Authors:

The manuscript by Hu et al has presented a new separator that allows robust operation at low film thickness and thereby improves the frequency response of electrochemical capacitors. Majority of data and analysis support their claims well. There are some points below to clarify:

Reply:

Thank you for carefully and professionally reviewing our Manuscript, acknowledging our research, and providing valuable suggestions. We have made corresponding revisions in the revised Manuscript. Please check the revised parts in red font. The following are the point-to-point replies to the comments.

1. *We agree that reducing separator thickness is important, the area of the capacitors will also reduce series resistance (resistors in parallel) in a cell. For example, in commercial devices with larger areas, the ESR naturally decreases due to the larger surface area. More discussion is needed to explain the effect of thin separators when using for large area devices. For fair comparison, it is crucial to provide the device area (before normalization) and compares to the total electrode area in AEC.*

Reply:

Thanks for the valuable suggestion. We have added discussion and materials in the revised Manuscript and the Supplementary Information file.

As you mentioned, commercial aluminum electrolytic capacitors usually adopt a winding structure to extend the electrode area in a confined footprint (Fig. R11). This structure is actually a parallel integration form, thus reducing ESR in a limited footprint area. Similarly, the stack integration in this manuscript is another parallel integration form, which uses stacks of units but continuous winding.

As mentioned, a fair comparison should be made between unit to unit or device to device. Therefore, we herein disassembled a commercial aluminum electrolytic capacitor and calculated its areal capacitance (Fig. R11 c). The electrode area is approximately $1\text{ cm} \times 35\text{ cm} = 35\text{ cm}^2$, and the footprint area of the device is $0.5\text{ cm} \times 0.5\text{ cm} \times \pi = 0.78\text{ cm}^2$. The areal capacitance at 120 Hz of the commercial aluminum electrolytic capacitor is 2.3 mF cm^{-2} normalized by the device and 0.081 mF cm^{-2} normalized by the material.

For comparison, the area of the device and electrode of TAS-LFEC are the same, $0.5\text{ cm} \times 0.5\text{ cm} = 0.04\text{ cm}^2$, while the stack number varies for different devices (Fig. R11 d). The areal capacitance at 120 Hz of the TAS-LFEC is 65.6 mF cm^{-2} normalized by a (10-layer) device and 4.8 mF cm^{-2} normalized by the material.

In Fig. 3d, we compared the two using values normalized by the material; In Fig. 5d, we used values normalized by the device.

Fig. R11 | **a,b**, The optical images of a commercial aluminum electrolytic capacitor with a capacitance of 3.3 mF. **c**, Areal capacitance of the commercial aluminum electrolytic capacitor normalized by device or material. **d**, The optical images of the TAS-LFEC. **e**, Areal capacitance of TAS-LFECs normalized by device or material.

2. Analyzing self-discharge performance is crucial for this manuscript due to the ultra-thin $3\ \mu\text{m}$ separator. In Figure S18, is y-axis unit really A cm^{-2} because that is quite high. Additionally, performing a Coulombic efficiency analysis at different current densities is necessary to better understand potential energy losses associated with the thin separator. A comparison of leakage current and self-discharge performance with a commercial separator is also needed.

Reply:

Thanks for your valuable advice. We enriched this part and added supplementary materials to the revised Manuscript.

The y-axis uses A cm^{-2} because the charge-discharge current density at normal operation is at this level and to clearly show a broad range. As shown in Fig. R12 b, the working-mode current density at line filtering is around 100 mA cm^{-2} , far larger than the device's leakage current density, indicating its little impact on the line filtering performance.

We obtained the coulombic efficiency of TAS-LFEC with charging and discharge current density varying from 4.8 mA cm^{-2} to 4.8 A cm^{-2} (Fig. R12 a). Although little fluctuation is observed at different current densities due to the equipment noises, coulombic efficiencies are mostly higher than 99%.

In addition, TAS has a similar leakage current and self-discharge behavior to the commercial separator, even slightly better. The origin of the leakage could be the instability of electrode materials or electrolytes. However, from the comparison between the self-discharge voltage signal and the working-mode voltage signal, one can conclude that the self-discharge has little impact on the normal line filtering function.

Fig. R12 | **a**, Coulombic efficiency of TAS-LFEC at varying current densities. **b**, Comparison of leakage current of TAS and commercial separator at a bias voltage of 1 V. The green dashed line indicates the working-mode current density. **c**, Comparison of the self-discharge behaviors between TAS and commercial separator. **d**, Comparison between the self-discharge voltage signal and the working-mode voltage signal. The green dashed line indicates the working-mode voltage signal.

3. In this paper, the ESR of the devices is below 0.05 ohms, approaching the electrical resistance of the conductive wires in the characterization equipment. It's important to describe the method used to mitigate the influence of resistance from the conductive wires. Is that the reason why Au current collectors were used? But Au is impractical for commercial applications, so it would be useful to explain the rationale of using Au.

Reply:

Please see our test equipment in Fig. R13 b. Several parts could contribute to the overall internal

resistance during the test, including the test wire, the probe's resistance, contact resistance, the resistance of the current collector (the referred Au electrode), the resistance of electrode materials, and the separator's resistance. The contributions of those parts are summarized in Fig. R13 a.

Typically, a tested unit has an area of $0.2 \times 0.2 \text{ cm} = 0.04 \text{ cm}^2$, the series resistance at 120 Hz is $36 \text{ m}\Omega \text{ cm}^2$, and the series resistance during the test is 0.9Ω . The Au-Au current collector accounts for a small portion of $10 \text{ m}\Omega$, and this value will decrease with integration in parallel. Instead, the resistance of the electric wires, probes, and contact should be subtracted from the system, because they are not a part of the device. This can be easily realized because they are linear elements being connected in the test equipment in series.

The reasons why we adopted Au electrode are: 1) Au has good chemical inertness. We can avoid side reactions when we discuss the mechanisms; 2) Au has good ductility, the Au pieces here are only $7 \mu\text{m}$ -thick. Adopting them can save a lot of space in the integrated device. We have tried to use graphite foil as the current collector. They perform similarly but are much thicker than Au pieces ($30 \mu\text{m}$ vs. $7 \mu\text{m}$). Maybe in occasions that don't need extremely high volumetric capacitance, we can adopt graphite foils as current collectors in the next-step applications.

Fig. R13 | **a**, Resistance constitution of the wholistic test equipment. **b**, Optical and schematic diagram of the test equipment.

4. In Fig 5g, we need more explanation on the comparison between Al electrolytic capacitor (AEC) and the capacitor (TAS-LFEC) in this work. The authors mentioned that both capacitors have the same capacitance (also please add that specific value in the caption), and the AEC and TAS-LFEC had the same phase angle in Fig 3c. Then RC constant should remain the same, since load was 10 Ohm. Please elaborate on the reason the AEC fail to rectify if the phase angle and RC constant are the same as TAS-LFEC?

Reply:

Thanks for the suggestions. The specific capacitances are added in the captions in the revised Manuscript.

We are sorry for the misleadingness. In Fig. 5g, the electrolytic capacitors are chosen to have nearly the same volume as the TAS-LFECs we used. The capacitance of the electrolytic capacitor is 10 μ F, and the capacitance of the 20-layer stack-integrated TAS-LFECs is 6.3 mF. The reason the electrolytic capacitor failed in filtering is that its capacitance was much lower than needed. During filtering, a

small part of the alternative signal passes through the capacitor, and a large part remains in the output signal, leading to an extensive ripple voltage in the output signals.

5. Please comment on the separator uniformity (edge vs center). Has the separator been tested with electrolytes with alkaline pH?

Reply:

To test the separator uniformity, we tested different locations in a piece of the separator and indexed them from 1 to 7 (Fig. R14). The statistical analysis results are: 1) The tested TAS separator has an average thickness of 3.2 μm , with a mean deviation of 0.1 μm . 2) The tested TAS separator has an average ionic resistance of 29 $\text{m}\Omega \text{ cm}^2$, with a mean deviation of 1.6 $\text{m}\Omega \text{ cm}^2$. These results are also added to the revised Supplementary Information file. Basically, the uniformity is acceptable for the stack integration of TAS-LFECs.

Fig. R14 | Uniformity test of the TAS separator. **a**, The optical image of the TAS separator. The rectangles indicate the locations where samples are taken. **b**, Nyquist diagrams of the tested samples. **c**, Thicknesses of the tested samples. **d**, Ionic resistance of the tested samples.

In addition, the electrochemical performances are tested in an alkaline electrolyte (3 M KOH). As shown in Fig. R15, the phase angle at 120 Hz is -68° , and the areal capacitance is 5.7 mF cm^{-2} . This performance degradation can be ascribed to the increase in ionic resistance to $80 \text{ m}\Omega \text{ cm}^2$.

Fig. R15 | Electrochemical performances of TAS-LFEC in an alkaline electrolyte (3 M KOH). **a**, Bode diagram of TAS-LFEC in 3 M KOH electrolyte. **b**, Plot of areal capacitance versus frequency of TAS-LFEC in 3 M KOH electrolyte. **c**, Nyquist diagram of TAS-LFEC in 3 M KOH electrolyte.

6. *Did the authors evaluate separator performance with electrodes with thick/high loading, with a loading range similar to those of commercial supercapacitors? At what electrode thickness/loading will the separator effect become negligible as resistance is dominated by electrodes?*

Reply:

As shown in Fig. R16 f, the separator dominates the overall series resistance of the TAS-LFEC at the range of lesser electrode material thickness. It is observed that when the electrode material thickness is higher than approximately 18 μm , the resistance of the electrode materials (including the electronic resistance and ionic resistance) plays a leading role.

As an analysis, this phenomenon may originate from three factors when the electrode materials get thicker: 1) The electronic resistance increases; 2) The ionic transport path becomes more distorted and longer; 3) The capacitance gets higher, and therefore, together with the above two factors, renders the result that the typical frequency response moves to the low-frequency region. These hypotheses can be verified by evolution in Nyquist diagrams. As shown in Fig. R16 b and c, the curves get more and more distorted at the high-frequency region, indicating that the ionic diffusion is getting more hindered. At higher electrode material loads, a new impedance arc emerges at the high-frequency region, because the key capacitor-behavior slows and moves to the low-frequency region, as a result of the above three factors.

Fig. R16 | Electrochemical performances of TAS-LFECs with varying electrode material thickness. **a**, Schematic diagram of TAS-LFECs with varying electrode material thickness. **b,c**, Nyquist diagrams of TAS-LFECs with varying electrode material thickness. Red arrows indicate the tendency of impedances as the electrode material thickness increases. **d**, Bode diagrams of TAS-LFECs with varying electrode material thickness. **e**, Plot of C_A with respect to frequency of TAS-LFECs with varying electrode material thickness. **f**, Histogram of R_s and SR at 120 Hz of TAS-LFECs with varying electrode material thickness.

7. As a future design rule, please comment on the design rule for the separator, such as the ratio of GO to cellulose content to optimize the pore size. What is the optimal GO size so that it can acts an anchor while keeping pore size large enough for ion transport.

Reply:

The design rule for the separator should have two basic criteria: 1) The separator should have good reliability that can steadily separate the positive and negative electrodes at normal operation, and 2)

The separator should have enough porous structure to store electrolyte and for the fast transport of ions.

As for the ratio of CNF to GO, we tested the TAS with varying CNF to GO weight ratios from 1:0 to

2:1. The results are presented in Fig. R 17. Basically, the introduction of GO will worsen the ionic

resistance of the pristine CNF membrane. With a small amount of GO addition (20:1 or 10:1), the

increment of ionic resistance is negligible, while a large amount of addition leads to a large increase in ionic resistance. This may be attributed to the less shielding on pores at little amount of addition.

Fig. R17 | Ionic resistance of TAS with different CNF to GO weight ratios. **a**, Nyquist diagram of TAS with different CNF to GO weight ratios. **b**, Ionic resistances of TAS with different CNF to GO weight ratios.

As for the GO size, it is hard to precisely control the GO size. Therefore, we chose three typical conditions (no addition of GO and addition of GO with sizes of 400 nm and 10 μm) for comparison. Given that the pore's sizes are around one micrometer, GO sheets with sizes smaller than one micrometer could avoid shielding pores. A distinct difference in ionic resistance is observed between the sheet sizes of 400 nm and 10 μm (Fig. R18). On the other hand, the ionic resistance remains nearly the same after the addition of GO sheets with a size of 400 nm, indicating that the shielding for pores can be neglected. Therefore, we anticipated that the optimal GO sheet size should be smaller than one micrometer.

In conclusion, the thread-anchor structure of CNF and GO has the optimal performance (tensile strength of 156 Mpa, and ionic resistance of 25 $\text{m}\Omega \text{ cm}^2$) with the parameters being set as follows: GO sheet size is around 400 nm and weight ratio of CNF to GO is around 10:1.

Fig. R18 | Investigation of the influences of GO sheet sizes on ionic resistances. **a**, Schematic diagrams of TAS (with GO sheet size of ~400 nm) and CNF-LGO membrane (with GO sheet size of ~10 μm). **b**, Nyquist diagrams of pristine CNF membrane-based LFEC, TAS-LFEC, and CNF-LGO membrane-based LFEC.

Editing issues:

8. In Fig 1b, the box plot was described in the text that the separator accounts for 60% of the total series resistance. This message is important and it can be observed that the ionic resistance of electrodes and separators are similar. However, how was this exact percentage calculated, when the data were quite spread out? Was it from comparing the median lines?

Reply:

Thank you so much for this comment. This value is not precise. We have corrected it to 54% in the revised Manuscript. The calculation process is as follows.

We have referenced 12 articles concerning the line-filtering electrochemical capacitors. The 60% percentage come from the equation, $\frac{\bar{R}_s}{\bar{R}_s + \bar{R}_m}$, where \bar{R}_s is the average separator ionic resistance, and \bar{R}_m is the average electrode material ionic resistance at 120 Hz. However, following the reviewer's advice, we found this number varies much in different articles as the systems are quite different.

Therefore, we recalculated $\frac{R_s}{R_s + R_m}$ for each referenced article. And the average number, $\frac{\bar{R}_s}{\bar{R}_s + \bar{R}_m}$, is 54%.

9. The X and Y axes in all EIS results within this manuscript use different scales, which should be corrected for consistency. The EIS results of TAS and CNF-LGO in Figure 4d display distinct behaviors at high frequencies; further explanation is needed to clarify these differences and how they relate to the separator design. In Figure 5b and 5c, please explain the appearance of bumps in the 10-100 Hz region.

Reply:

Thanks for the suggestion. Normally, EIS should keep X and Y axes scales the same. We tried our best to follow this norm and changed some figures in the revised Manuscript. However, we have to rescale the axes to show all the key impedance points in some figures. For example, the normal figure of Fig. 4d loses details at the high-frequency region and is unclear enough for readers (Fig. R 19 here).

Fig. R19 | Nyquist diagrams of TAS-LFEC and CNF-LGO membrane-based LFEC at different axes scales.

The distortions at the high-frequency region in Fig. 4d may reflect the ionic diffusion behaviors of the two devices. TAS-LFEC has a straighter Nyquist curve than CNF-LGO membrane-based LFEC. We reason that the distortion originates from the ionic diffusion limitation in CNF-LGO membrane-based LFEC, as the pores are shielded by the LGO sheets.

In Fig. 5b, the bumps come from the noises of the test equipment (CHI 660e), because the sensitivity

changes with the tested range and shows some fluctuation.

10. *In Figure 5d, please indicate how many layers were used in the cell structures for all the cited papers.*

Reply:

Thank you for the suggestion. The cited papers all have only one device unit, which has one positive electrode and one negative electrode. As far as we know, this Manuscript should be the first paper to stack integrate LFECs in parallel.

11. *In Figure S19, please specify how long the cycling test took to complete. Additional details, including the device's capacitance, GCD curves before and after cycling, and EIS results, are necessary to support the cycling stability conclusions.*

Reply:

Thanks for the suggestion. The details of the cyclic test are added in the revised Supplementary Information file.

The cyclic test applies a galvanostatic charge and discharge test on the device with a current density of 100 mA cm^{-2} . An electrochemical impedance spectroscopy is conducted every 1,000 cycles. This repeated test unit took about 5 minutes. Counting all the quiescent times, the overall test for one million's cyclic test requires 3 days to complete.

As shown in Fig. R20, TAS-LFEC can sustain over one million cycles. The tested device shows good stability in areal capacitance and phase angle, the retentions are nearly 100%. The electrochemical impedance spectroscopy results remain the same at the end of the test. In addition, the Coulombic efficiency remains higher than 99% during the wholistic cyclic test.

Fig. R20 | Details of cyclic test of TAS-LFEC. a Areal capacitance at 120 Hz and its retention during the cyclic test. b, Phase angle at 120 Hz and its retention during the cyclic test. c, Nyquist diagrams of TAS-LFEC at the start and end of the cyclic test. d, Galvanostatic charge and discharge test curves of TAS-LFEC at the start and end of the cyclic test.

12. *In Table S2, averaging data from different papers has no clear physical meaning and should be reconsidered.*

Reply:

Thanks for the advice, those values are deleted in the revised Supplementary Information file.

13. *The fonts in all the plots are too small and not legible. Please increase the font size to make them readable without the need to magnify.*

Reply:

We have increased the font size as much as possible under the editorial requirement.

14. *The grammar in the title needs correction. Please edit “ultra-highly” and “high-load filter”. Perhaps the authors mean “Separator with ultrahigh ionic conductivity in electrochemical supercapacitors for ac line filtering (at high power)”. There are other instances where the text needs to improve; please check. And perhaps add in the introduction that “line-filtering” means alternating current (ac) line filtering, even though it might be obvious to us in the field, it is still better to clarify it as ac line filtering.*

Reply:

Thanks for the advice. We then changed the title to, **Separator with ultrahigh ionic conductivity enables electrochemical capacitors to line-filter at high power.**

An introduction to AC line filtering was added to the Introduction section of the revised Manuscript.

We also scrutinized the Manuscript to reduce syntax errors.

Reviewer #4 (Remarks to the Author):

Reply:

Thank you for your distinguished review work, which, along with other reviewers' suggestions, greatly improved the Manuscript.

REVIEWERS' COMMENTS

Reviewer #1 (Remarks to the Author):

The authors have adequately addressed my previous comments, and the manuscript is now suitable for acceptance. Below are a few minor comments for further improvement:

Reply:

Thank you for your valuable advice. Through our communication with you, we have learned and noticed many things that can help us improve our manuscript. The following are the responses to the comments, and corrections in the revised manuscript are marked in red font.

Line 332 and 334: The unit $Fcm^{-3}V^2$ is used for capacitance density, which is scientifically incorrect. While it is understood that achieving high rated voltage requires connecting multiple devices in series—resulting in a volumetric capacitance decrease proportional to the square of the voltage—such a unit is not appropriate.

Reply:

Thank you for this point. We corrected the expression in the revised manuscript.

Figure 5g: The TAS-EC device is reported to have dimensions of 2mm x 2 mm x 26.4 μm × 10. Could the authors confirm whether the AEC device has similar dimensions?

Reply:

For the dimensions, the commercial aluminum electrolytic capacitor doesn't have such a small size. Therefore, we chose a substituent one which has the equivalent capacitance: The dimension of the 10-layer stack-integrated LFEC is $0.2\text{ cm} \times 0.2\text{ cm} \times 0.03\text{ cm} = 1.2 \times 10^{-3}\text{ cm}^3$ (due to the non-ideal stack technique, the thickness is about 0.05 cm). We assume a commercial aluminum electrolytic capacitor has a volumetric capacitance of $4 \times 10^{-3}\text{ F cm}^{-3}$, which is calculated from a polymer aluminum electrolytic capacitor (1000 μF , 2 V, $\phi 6.3\text{ cm} \times 8\text{ cm}$). In this way, the same-volume aluminum electrolytic capacitor should have a capacitance of 4.8 μF . To exclude the underestimation of electrolytic capacitors' volumetric capacitance, we chose an electrolytic capacitor with a capacitance of 10 μF as the same-volume aluminum electrolytic capacitor.

Additionally, it is noted that the maximum applied voltage for TAS-EC is 1.5 V, not the rated 1 V as previously indicated.

Reply:

Thank you for the correction. We used 1 V as the rated voltage in the calculation to leave a margin for the maximum withstand voltage.

"Built-in Voltage": The term "built-in voltage" has been used in a manner that differs from the conventional definition (e.g., in semiconductor physics). Can the authors clarify its specific meaning within the context of this manuscript?

Reply:

Thank you for the reminder. We have added the definition in the revised manuscript.

Reviewer #2 (Remarks to the Author):

The revisions make this manuscript suitable for publication in Nature Communications.

Reply:

Thank you for your recognition and instructions, which improved our manuscript significantly.

Reviewer #3 (Remarks to the Author):

This revision has satisfactorily addressed the reviewers' comments.

Reply:

Thank you for your valuable suggestions and recognition of our manuscript.

Reviewer #4 (Remarks to the Author):

I co-reviewed this manuscript with one of the reviewers who provided the listed reports.

This is part of the Nature Communications initiative to facilitate training in peer review and to provide appropriate recognition for Early Career Researchers who co-review manuscripts.

Reply:

Thank you for your excellent review work.